# RanGTP and the actin cytoskeleton keep paternal and maternal chromosomes apart during fertilization

Masashi Mori[1,2] , Tatsuma Yao[3,4] , Tappei Mishina[2], Hiromi Endoh[5], Masahito Tanaka[6], Nao Yonezawa[4], Yuta Shimamoto[6] , Shigenobu Yonemura[5,7], Kazuo Yamagata[4], Tomoya S. Kitajima[2], and Masahito Ikawa[1]

Zygotes require two accurate sets of parental chromosomes, one each from the mother and the father, to undergo normal embryogenesis. However, upon egg–sperm fusion in vertebrates, the zygote has three sets of chromosomes, one from the sperm and two from the egg. The zygote therefore eliminates one set of maternal chromosomes (but not the paternal chromosomes) into the polar body through meiosis, but how the paternal chromosomes are protected from maternal meiosis has been unclear. Here we report that RanGTP and F-actin dynamics prevent egg–sperm fusion in proximity to maternal chromosomes. RanGTP prevents the localization of Juno and CD9, egg membrane proteins that mediate sperm fusion, at the cell surface in proximity to maternal chromosomes. Following egg–sperm fusion, F-actin keeps paternal chromosomes away from maternal chromosomes. Disruption of these mechanisms causes the elimination of paternal chromosomes during maternal meiosis. This study reveals a novel critical mechanism that prevents aneuploidy in zygotes.

## Introduction

A new life begins with fertilization, which fuses an egg with a sperm to produce a zygote. In all animals examined thus far, spermatogenesis involves the completion of two meiotic divisions (meiosis I and meiosis II) that produce mature sperm, each carrying one set of paternal chromosomes. In contrast, in vertebrates including humans, oogenesis arrests before completion of the second meiotic division (meiosis II), which results in fully matured eggs each carrying two sets of maternal chromosomes. Thus, upon fertilization, the zygote has three sets of chromosomes, one from the sperm and two from the egg. Fertilization resumes maternal meiosis II, which eliminates one set of maternal chromosomes, but not paternal chromosomes. The zygote thereby becomes diploid, containing both parental genomes, which is a prerequisite for undergoing normal embryogenesis. If paternal chromosomes are eliminated by maternal meiosis, the zygote becomes parthenogenetic or aneuploid, which leads to pregnancy loss and congenital diseases. However, mechanisms that prevent paternal chromosomes from being eliminated into the polar body by maternal meiosis remain poorly understood.

One mechanism to protect paternal chromosomes is to keep them away from the maternal spindle during maternal meiosis. In *Caenorhabditis elegans*, an F-actin–dependent mechanism prevents interactions between paternal chromosomes and the

meiotic spindle following fertilization (Panzica et al., 2017; Panzica and McNally, 2018). Before these postfertilization mechanisms, a spatial bias in sperm entry sites on the egg surface may facilitate the initial positioning of paternal chromosomes away from the maternal spindle. In mouse eggs, EM has thus far identified two distinct regions on the egg surface: (1) a region covered with microvilli, finger-like membrane protrusions containing dense F-actin bundles (Nicosia et al., 1977; Dalo et al., 2008; Mackenzie et al., 2016; Uraji et al., 2018), and (2) a microvillus-free "smooth" region, which largely overlaps with the actin cap, an F-actin–rich cortical domain surrounding the maternal spindle (Maro et al., 1984 and 1986; Longo and Chen, 1985; Deng et al., 2007; Dehapiot et al., 2013). Since egg–sperm fusion does not occur or seldom occurs in the microvillus-free region where the maternal spindle is located (Johnson et al., 1975; Nicosia et al., 1977; Yanagimachi, 1988), paternal chromosomes are localized away from maternal chromosomes immediately after fusion in zygotes.

Because sperm is often observed with microvilli on the egg surface in mammals (Yanagimachi and Noda, 1970; Shalgi and Phillips, 1980), it has been speculated that sperm fusion is promoted by microvilli and/or is blocked by the actin cap. However, the disruption of microvilli and the actin cap by treatment with

[1]Research Institute for Microbial Diseases, Osaka University, Osaka, Japan;   [2]Laboratory for Chromosome Segregation, RIKEN Center for Biosystems Dynamics Research, Kobe, Japan;   [3]Research and Development Center, Fuso Pharmaceutical Industries, Ltd., Osaka, Japan;   [4]Graduate School of Biology-Oriented Science and Technology, Kindai University, Wakayama, Japan;   [5]Laboratory for Ultrastructural Research, RIKEN Center for Biosystems Dynamics Research, Kobe, Japan;   [6]Physics and Cell Biology Laboratory, National Institute of Genetics & Department of Genetics, SOKENDAI University, Kanagawa, Japan;   [7]Department of Cell Biology, Tokushima University Graduate School of Medical Science, Tokushima, Japan.

Correspondence to Masahito Ikawa: ikawa@biken.osaka-u.ac.jp;   Masashi Mori: masashi.mori@riken.jp.

F-actin inhibitors does not block sperm fusion (McAvey et al., 2002; Runge et al., 2007), indicating that these F-actin–based cell surface domains are dispensable for sperm fusion. Instead, recent studies using gene knockout mice have identified two egg membrane proteins that are essential for sperm fusion: Juno and CD9. Juno directly interacts with the sperm membrane protein IZUMO1 (Inoue et al., 2005; Bianchi et al., 2014; Ohto et al., 2016; Kato et al., 2016) while CD9 organizes the membrane to enhance sperm–egg binding (Le Naour et al., 2000, Miyado et al., 2000, Kaji et al., 2000) and is known to localize at microvilli (Runge et al., 2007; Miyado et al., 2008). However, functional relationships of F-actin–based cell surface domains with Juno and CD9 remain to be investigated.

Understanding of intracellular dynamics during fertilization and maternal meiosis in zygotes has been limited due to the lack of robust techniques for high-resolution live imaging in co-cultures of eggs and sperm (Piotrowska and Zernicka-Goetz, 2001; Motosugi et al., 2006; Jin et al., 2011). In this study, we established a live-imaging technique that enabled us to track maternal and paternal chromosomes from shortly after egg–sperm fusion until the exit of maternal meiosis. The results demonstrate that sperm fusion sites are spatially biased to cell surface regions distal to maternal chromosomes, which is determined by maternal chromosome-mediated RanGTP signaling through F-actin–dependent and –independent pathways. The RanGTP signaling prevents the localization of Juno and CD9 at the cell surface region in proximity to maternal chromosomes. We identify a novel ring-like cell surface domain enriched with lamellipodia-like protrusions, which move together with sperm on the egg surface toward regions distal to maternal chromosomes before their fusion. The RanGTP- and F-actin–dependent spatial bias in egg–sperm fusion sites facilitates the initial positioning of paternal chromosomes away from the maternal spindle. Moreover, following egg–sperm fusion, paternal chromosomes are kept at a distance from the maternal spindle in an F-actin–dependent manner. Forced localization of paternal chromosomes into close proximity to maternal chromosomes through intracytoplasmic sperm injection (ICSI) causes the elimination of paternal chromosomes during maternal meiosis.

## Results

### Paternal chromosomes are spatially separated from maternal chromosomes

Live imaging of chromosome dynamics from the time of egg–sperm fusion until the exit of maternal meiosis in zygotes has been hampered by a lack of methods to maintain eggs at stable positions in cocultures with actively moving sperm and to label paternal chromosomes devoid of histones immediately after egg–sperm fusion. To overcome these difficulties, we used BSA-free medium to attach eggs to a coverglass and then added BSA to enable one-to-one egg–sperm fusion. Furthermore, we found that mCherry-tagged methyl CpG binding domain (MBD; Yamagata et al., 2005; Yamagata, 2010), which directly binds to DNA, labeled paternal chromosomes immediately after egg–sperm fusion (Fig. S1 A and Video 1).

These techniques enabled us to robustly determine egg–sperm fusion sites by live imaging (Fig. 1 A and Video 2). We found that egg–sperm fusion occurred at an angle of >40° and >24 μm from maternal chromosomes (Figs. 1 B and S1 B). The distribution of egg–sperm fusion sites was compared using simulations where sperm fuse randomly anywhere on the egg surface (we assumed that the egg is spherical, Fig. 1 B). The experimental data showed that egg–sperm fusion sites were significantly biased to regions distal to maternal chromosomes (Figs. 1 B and S1 C). These findings suggest that egg–sperm fusion is unfavored at cell surface regions in proximity to maternal chromosomes.

Moreover, our newly developed live-imaging techniques allowed us to track chromosomes in 3D following egg–sperm fusion. That analysis revealed that paternal chromosomes moved dynamically along the periphery of the zygote (Figs. 1 C and S1 C). During this period, the displacement of paternal chromosomes reached a maximum distance of ≤43 μm (Fig. 1 D), which was greater than that of the meiotic spindle carrying maternal chromosomes (Fig. S2 B). Interestingly, despite the dynamic and long-distance nature of paternal chromosome movements, they never reentered the 30-μm region surrounding maternal chromosomes (Fig. 1 E). In rare cases where egg–sperm fusion occurred within the 30-μm region surrounding maternal chromosomes, the paternal chromosomes rapidly moved out of that region (Fig. 1, C and E, arrowheads). To check if the movement of paternal chromosomes is directional or random, we calculated the angular displacement of paternal chromosomes toward maternal chromosomes at every time point between sperm fusion and 100 min after anaphase onset. Since the directionality was not obvious, we tested four models using two variables to explain the movement of paternal chromosomes: the angular displacement has a correlation with the position of paternal chromosomes (angle only), with the time after sperm fusion (time only), with both of those variables (angle + time), and with both of those variables and their interaction (angle × time; Fig. 1 F, left, and Fig. S2 A). These models were evaluated based on Akaike's information criterion (AIC) values, and the most likely model was angle × time (AIC: angle only, 3,078.1; time only, 3,047.9; angle + time, 3,040.1; angle × time, 3,014.8). According to the prediction of the angle × time model, we separated the data into the spindle half and the opposite half of zygotes and found that paternal chromosomes moved away from maternal chromosomes in the spindle half during an early phase of the fertilization process (Fig. 1 F, right). These findings suggest that zygotes actively keep paternal chromosomes at a distance from maternal chromosomes after sperm fusion.

In summary, these results suggest that zygotes spatially separate paternal chromosomes from maternal chromosomes by (1) biasing egg–sperm fusion sites to cell surface regions distal to maternal chromosomes, and (2) keeping paternal and maternal chromosomes apart following egg–sperm fusion.

### The localization of the egg–sperm fusion proteins Juno and CD9 corresponds to three segments of the cell surface

We next explored how egg–sperm fusion sites are biased to cell surface regions distal to maternal chromosomes. First, we

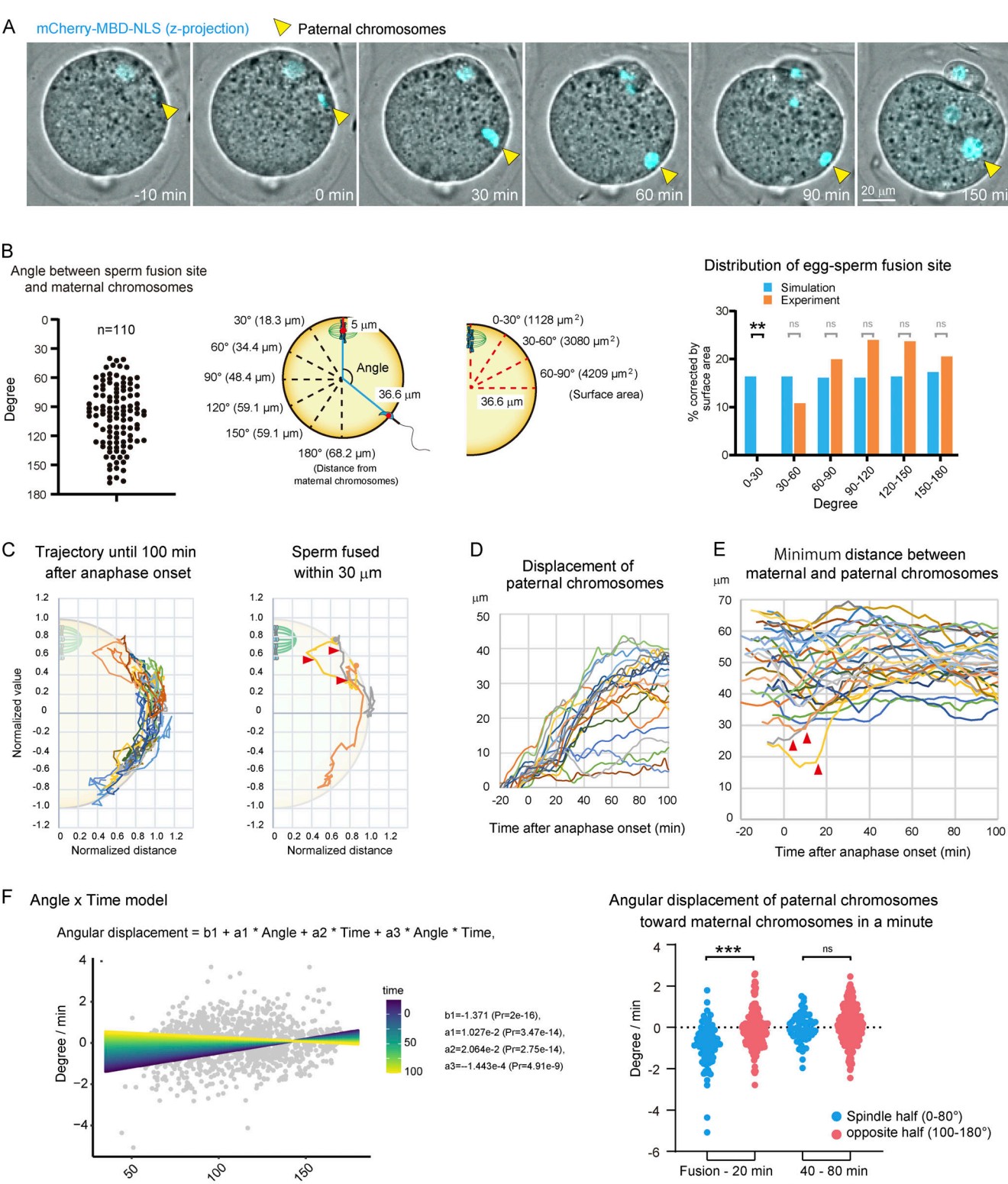

Figure 1. **Paternal chromosomes are spatially separated from maternal chromosomes during zygote meiosis. (A)** Live imaging of chromosomes in an egg expressing mCherry-MBD-NLS during the fertilization process. Maximum-intensity Z projection of the chromosome signal is shown. Time is relative to anaphase onset. A time-lapse video is shown in Video 2. The zona pellucida was softened by treatment with glutathione (A–D and F). **(B)** Left: Angle between maternal chromosomes and sperm fusion site. Middle: Parameters and assumptions for the simulation in which sperm fuse randomly on the egg surface. Right: Comparison between experimental data and a simulation where sperm fuse randomly anywhere on the egg surface. Data include zygotes that were poly-spermy (polyspermy rate was 40% of the fertilized eggs). Fisher's exact tests were used to obtain P values, and Holm correction was used for the correction of multiple comparisons (P value of 0–30° is $< 8 \times 10^{-3}$. P values of other areas are >0.05 [ns]). The rate of fusion events was corrected by the surface area after the statistical test. **(C)** Trajectories of chromosomes during the fertilization process. Paternal chromosomes move along the periphery of the zygote. Right:

Trajectory of paternal chromosomes, which fuse within 30 µm of maternal chromosomes. The values of the position of chromosomes were normalized by the length between the sperm fusion site and the center of the egg. Data include only zygotes that were monospermic (C–E). Arrowhead shows when paternal chromosomes rapidly moved out of the 30-µm region (C and E). **(D)** Displacement of paternal chromosomes from the sperm fusion site. **(E)** Distance between paternal and maternal chromosomes, which were closer after anaphase onset. **(F)** Left: Angle × time model in which the angular displacement has a correlation with the position of paternal chromosomes and the time from sperm fusion, with negative interactions of these variables. The time from sperm fusion is shown in the color scale. Right: Angular displacement of paternal chromosomes toward maternal chromosomes in the minute during sperm fusion to 20 and 40–80 min after anaphase onset. Welch's *t* test was used to obtain the P value (P value of fusion to 20 min is <0.001). **, $P < 0.01$; ***, $P < 0.001$.

addressed whether the zona pellucida, a solid layer surrounding eggs, has physical or structural properties that may contribute to the spatial bias in sperm fusion sites. We measured the stiffness and porosity of the zona pellucida and found them to be independent of the position of maternal chromosomes (Figs. S3 and S4). Next, we focused on the localization of Juno and CD9, two maternal membrane proteins essential for sperm fusion (Bianchi et al., 2014; Miyado et al., 2000). Immunostaining showed that Juno and CD9 had polarized localization patterns that defined three distinct segments of the cell surface (Fig. 2 A, z-projection). First, on the egg hemisphere opposite to maternal chromosomes, Juno and CD9 had spot-like signals that densely covered the cell surface (dense Juno/CD9 area [DA]). Second, next to the DA, we found a ring-like segment containing fewer numbers of discrete signals of Juno and CD9 per area, at angles of 43° ± 10° to 64° ± 10° with maternal chromosomes (fewer Juno/CD9 area [FA]; Fig. 2 B). Third, at the remaining cap-like segment in proximity to maternal chromosomes, Juno and CD9 had no detectable discrete signals (no Juno/CD9 area [NA]). Close inspection of the discrete signals with Airyscan imaging demonstrated that Juno and CD9 largely, although not completely, colocalized (Fig. 2 C). These results suggest that the density of Juno/CD9-containing discrete structures (hereinafter Juno/CD9 structures) are spatially regulated to mark cell surface regions competent for sperm fusion.

### The densities of Juno/CD9 structures correlate with cell surface protrusions

Previous reports showed that CD9 is bound to microvilli in oocytes (Runge et al., 2007; Miyado et al., 2008). We therefore hypothesized that the Juno/CD9 structures are associated with cell surface protrusions. Consistent with that hypothesis, EM demonstrated that the cortex can be categorized into three different segments based on the densities of protrusions (Fig. 3 A). A high density of finger-like membrane protrusions termed microvilli was observed on the cortex of the hemisphere of the egg (Fig. 3 B), which likely corresponded to the DA. Next to that region, we found a ring-like segment containing flatter protrusions at a relatively low density (Fig. 3 B), which likely corresponded to the FA. Almost no protrusions were found at the remaining cap-like segment (Fig. 3 B), likely corresponding to the NA. Thus, the densities of Juno/CD9 structures are consistent with those cell surface protrusions.

### The ring-like segment of the cell surface has lamellipodia-like protrusions and relocates Juno and CD9 to regions distal to maternal chromosomes

Given the finding of the ring-like segment at the cell surface, we investigated the shape and dynamics of protrusions in that segment. In contrast to canonical microvilli in the DA, the shapes of protrusions in the ring-like segment were thin and flat, like the shapes of lamellipodia in tissue culture cells (Fig. 3 B; Stradal et al., 2001). Lamellipodia are often characterized by dynamic turnover and mobility. Live imaging demonstrated that Juno/CD9 structures have dynamic properties. They appeared de novo in the ring-like segment and moved directionally toward the DA (Fig. 3 C and Video 3). In contrast, the Juno/CD9 structures were relatively stable in the DA (Fig. 3 C and Video 3). Since the Juno/CD9 structures move away from maternal chromosomes in the FA region, we hypothesized that sperm moves together with Juno/CD9 structures on the egg surface before their fusion. Monitoring of sperm motions with the sperm tail marker Su9-DsRed2 (Hasuwa et al., 2010) showed that sperm in the FA region moved away from maternal chromosomes before fusion, whereas sperm in the DA region exhibited no significant motions (Fig. 3, D and E; and Video 4). These results suggest that in the ring-like segment, Juno/CD9 are localized on lamellipodia-like protrusions, and that these structures dynamically move together with sperm to regions distal to maternal chromosomes before their fusion on the egg surface.

### F-actin contributes to but is not solely responsible for blocking the formation of Juno/CD9 structures in proximity to maternal chromosomes

We addressed the possibility that the spatial control of the localization of Juno and CD9 is mediated by F-actin, which is responsible for forming cell surface protrusions (Runge et al., 2007) and a cortical domain called the actin cap (Longo and Chen, 1985) at regions distal to and proximal to maternal chromosomes, respectively. We used Latrunculin B (LatB), which blocks the formation of cell surface protrusions (Runge et al., 2007) and the actin cap (Fig. S5 A). As expected, the actin intensity was significantly reduced at all regions in LatB-treated eggs (Fig. S5 A). The localizations of Juno and CD9 remained overall polarized on the cell surface, but the borders of Juno/CD9 structures at the cell surface were less clear (Fig. 4 A, whole and FA). Live imaging demonstrated that acute treatment with LatB rapidly disrupted the localization patterns of Juno and CD9 in the FA region within 15 min, which was followed by the translocation or de novo appearance of spot-like Juno/CD9 structures into the former FA region (Fig. 4 B and Video 5). These Juno/CD9 structures often reached positions where the NA/FA border is normally found in control eggs (Fig. 4 C, LatB_NA/DA), and a small population of the Juno/CD9 structures were localized even nearer to maternal chromosomes (Fig. 4 C, LatB_closest focus). The cell surface of LatB-treated eggs had long, bleb-like membrane structures containing strong Juno signals but quite weak CD9 signals (Fig. 4 A), possibly due to the deformation of sphere-shaped, Juno-positive, CD9-negative structures found on the cell surface of untreated control eggs (Fig. 2 C). These results suggest that F-actin contributes to but is not solely responsible for blocking the

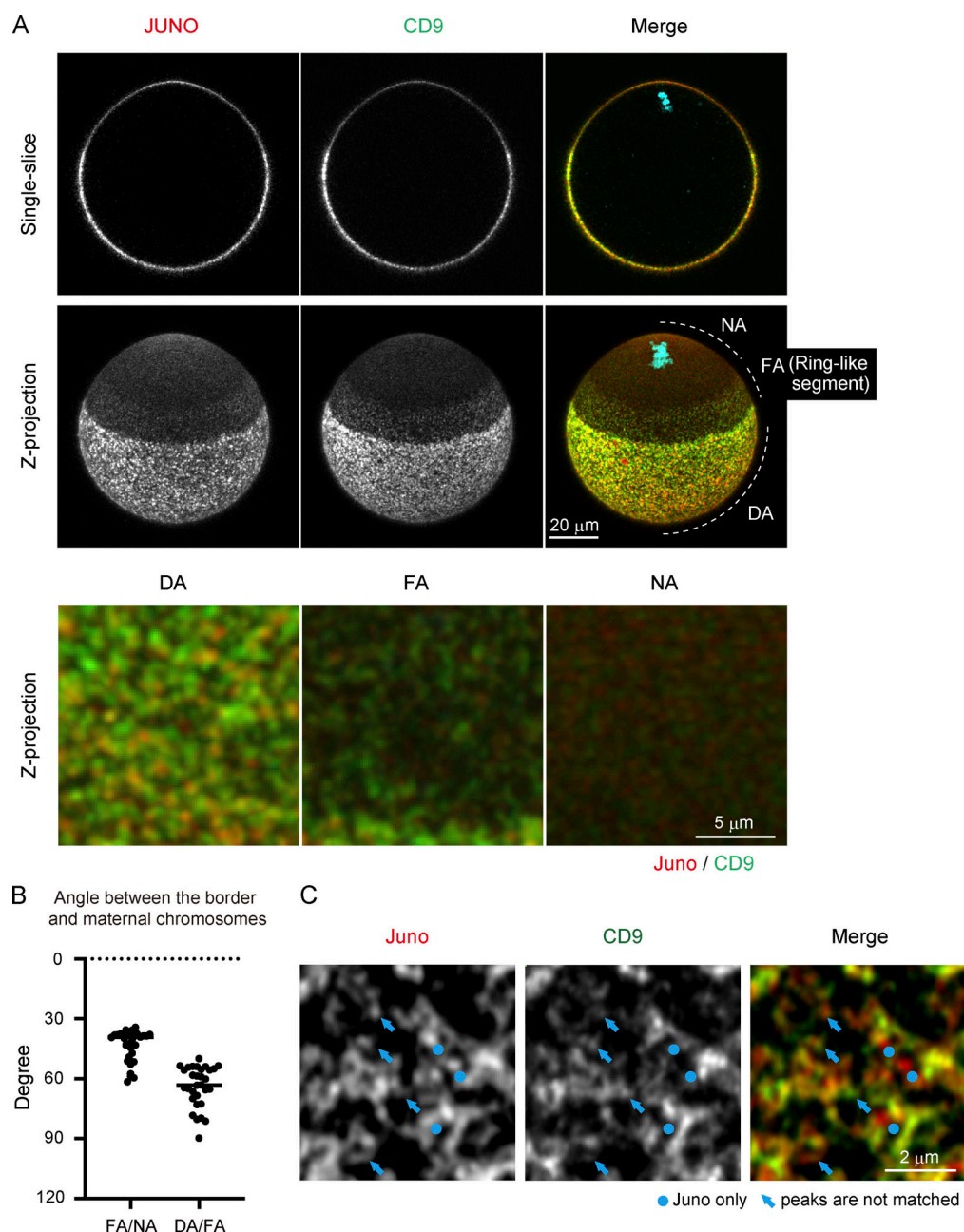

Figure 2. **Localization of Juno and CD9 defines three segments of the cell surface. (A)** Immunofluorescence of Juno and CD9. Single confocal section (single slice) and maximum-intensity Z projection (Z-projection) are shown. These membrane proteins form discrete structures and exhibit polarized localization patterns. The lower panel shows high-magnification images of DA (dense Juno/CD9 area), FA (fewer Juno/CD9 area), and NA (no Juno/CD9 area). The zona pellucida was removed (A–C). **(B)** Average angle between the borders and maternal chromosomes. The angles from three different positions on the border for each egg were averaged. **(C)** Superresolution image of Juno and CD9 by Airyscan detector. There are at least two populations: one contains Juno only (circles), and the other contains both Juno and CD9. In structures containing both Juno and CD9, the intensity peaks of Juno and CD9 are not always matched (arrows).

formation of Juno- and CD9-positive structures in proximity to maternal chromosomes.

**RanGTP activity surrounding maternal chromosomes prevents the localization of Juno and CD9**

We then characterized the F-actin–independent mechanism that prevents the localization of Juno and CD9 in proximity to maternal chromosomes. When chromosomes were dispersed in

eggs by treatment with nocodazole (Noc), we found that cell surface regions surrounding individual chromosomes excluded Juno and CD9 localization (Fig. 5 A). These observations suggest that chromosomes produce a diffusible signal that prevents Juno and CD9 localization. We therefore inhibited the activity of RanGTP (Carazo-Salas et al., 1999), a chromosome-mediated diffusible signal, by microinjecting the dominant-negative mutant RanT24N recombinant protein (Gruss et al., 2001; Nachury

## A

**Whole egg**

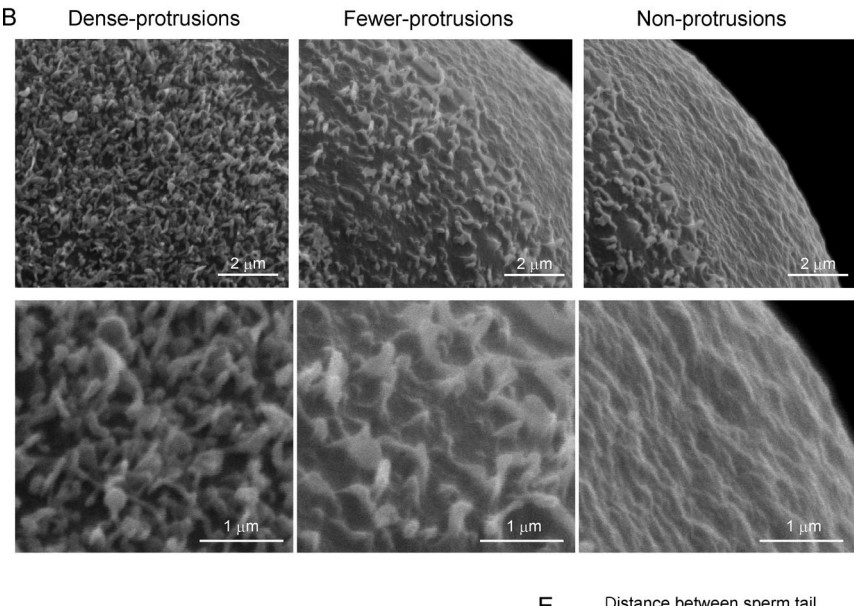

Non

(Ring-like segment)

Fewer

10 μm

Dense

## B

| Dense-protrusions | Fewer-protrusions | Non-protrusions |
|---|---|---|
| 2 μm | 2 μm | 2 μm |
| 1 μm | 1 μm | 1 μm |

**Figure 3. The densities of protrusions on the egg surface correlate with Juno/CD9 structures. (A)** Protrusions on the egg surface visualized by scanning electron microscopy (n = 21). The zona pellucida was removed in A–C by treatment with collagenase. **(B)** High-magnification images in dense-, fewer-, and nonprotrusion areas. In the dense-protrusion area, there are at least two types of structures: one is a finger-like shape, and the other is a round shape. In the fewer-protrusion area, the shape of protrusions is flat. **(C)** Live imaging of Juno/CD9 structures with fluorescent-labeled Juno and CD9 primary antibodies (n = 15, data of CD9 are not depicted). The Juno/CD9 structures in the FA region moved directionally toward the FA/DA border. The structures in the DA region are relatively stable. A time-lapse video is shown in Video 3. **(D)** Live imaging of chromosomes and sperm tails. Arrowheads indicate the initial binding position of sperm tails. Two holes were made in the zona pellucida near the actin cap on purpose in D and E. **(E)** The distance between the sperm tail and maternal chromosomes. Sperm that bind close to maternal chromosomes move away from maternal chromosomes before sperm fusion.

## C

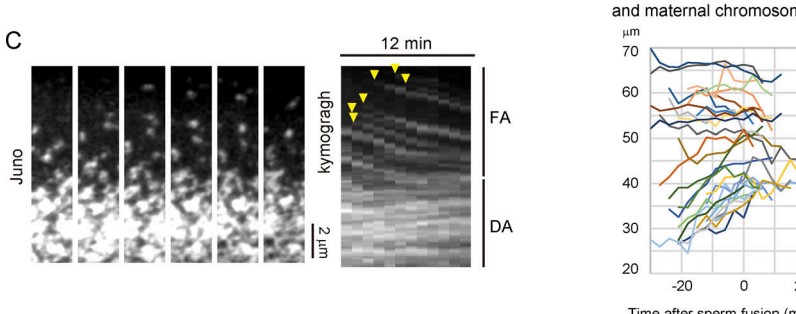

Juno

Kymograph

12 min

FA

DA

2 μm

## E

Distance between sperm tail and maternal chromosomes

μm

Time after sperm fusion (min)

## D

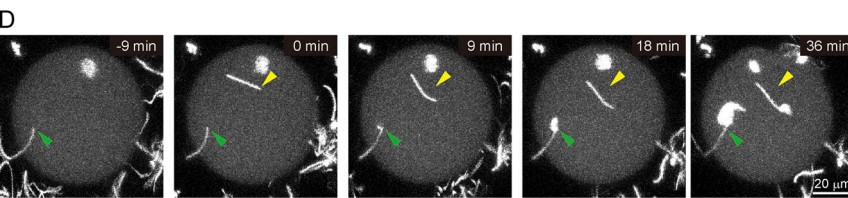

-9 min | 0 min | 9 min | 18 min | 36 min | 20 μm

Sperm tail (Su9-DsRed2) and chromosome (mCherry-MBD-NLS)

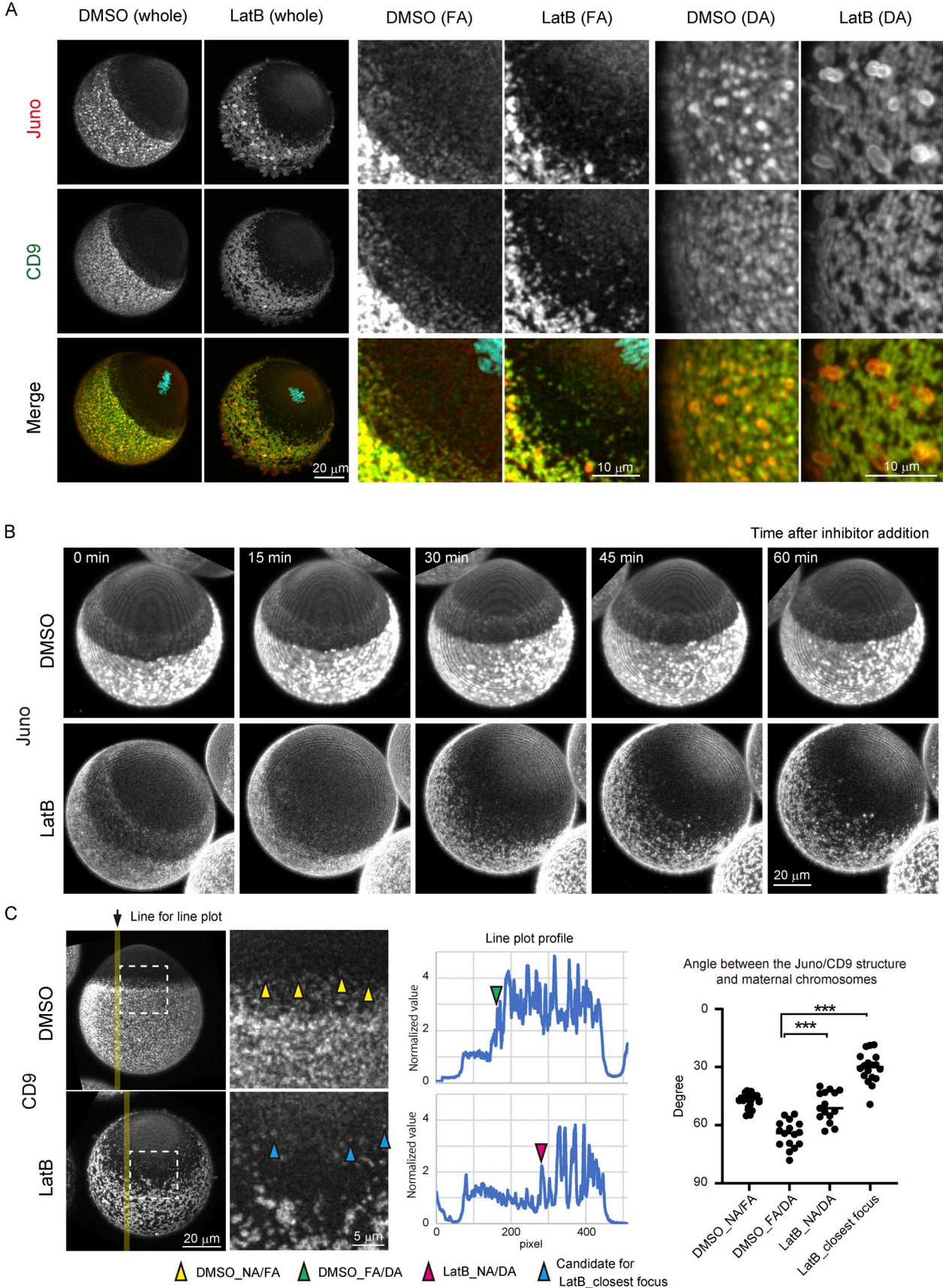

Figure 4. **F-actin contributes to but is not solely responsible for blocking the formation of Juno/CD9 structures around maternal chromosomes.** **(A)** Immunofluorescence images of Juno and CD9 following treatment with DMSO or LatB at low magnification (whole) and high magnification (FA and DA). In

LatB-treated eggs, the localization of Juno and CD9 remains overall polarized on the egg surface (whole). The FA/NA and DA/FA borders of Juno/CD9 structures become less clear in LatB-treated eggs (FA). Bleb-like long membrane protrusions containing strong Juno signals are observed in eggs treated with LatB (DA). The zona pellucida was removed (A–C). **(B)** Live imaging of Juno structures with a fluorescent-labeled Juno primary antibody in DMSO- or LatB-treated eggs. Juno structures in the FA region are disrupted rapidly, and Juno structures in the DA region are transferred into spot-like structures or new spot-like structures are formed de novo in the former FA region. Time-lapse video is shown in Video 5. **(C)** Angle between Juno/CD9 structures and maternal chromosomes after LatB treatment. The FA/DA borders in DMSO-treated eggs, and the NA/DA border in LatB-treated eggs, were determined as follows. We plotted the intensity of CD9 along a line on the maximum z-projection of image (yellow line), and the resultant plot (shown as Line plot profile) was used to acquire the position of local intensity increase (plot profile). The position was used to calculate the angle with the center of the egg in 3D. We acquired such angles from three lines per one oocyte and averaged them. For DMSO_NA/FA and LatB_closest focus, low-intensity foci of Juno/CD9 structure were manually selected (yellow and blue arrowhead). Welch's $t$ test was used to obtain the P value (P values between DMSO_DA and LatB_DA or LatB closest are both <0.001). ***, P < 0.001.

et al., 2001). We found that Ran-inhibited eggs had no detectable polarity of Juno and CD9 localization, while Juno and CD9 signals were found all over the cell surface, even at regions proximal to maternal chromosomes (Fig. 5 B). These effects were not fully attributable to disruption of the actin cap, a RanGTP-dependent domain of the cell surface surrounding maternal chromosomes, because disruption of the actin cap by treatment with LatB (Fig. S5 A) did not recapitulate the effects of RanGTP inhibition (Figs. 4 A and S5 B). These results suggest that chromosome-mediated RanGTP activity prevents the localization of Juno/CD9 structures in proximity to maternal chromosomes.

### Spatial bias in egg–sperm fusion is determined by RanGTP via F-actin–dependent and –independent pathways

Based on these findings, we addressed whether RanGTP- and/or F-actin–mediated pathways contribute to the spatial bias in egg–sperm fusion sites. In contrast to control eggs, where sperm fusion sites were significantly biased toward regions distal to maternal chromosomes, Ran-inhibited eggs had sperm fusion at positions all over the cell surface, including the region proximal to maternal chromosomes (Figs. 5 C and S5 C). LatB-treated eggs had a significantly lower spatial bias in sperm fusion compared with control eggs, although the vast majority of sperm fusion sites were still excluded from positions at angles of <30° with maternal chromosomes (Fig. 5 D). Note that sperm fusion sites in LatB-treated eggs appeared to be enriched around angles of 30°–60° rather than uniformly distributed across angles of 30°–180°, which may be due to the holes that were made in the zona pellucida near the perivitelline space to prevent polyspermy. Taken together with the fact that F-actin–mediated control of the cell surface is downstream of RanGTP (Deng et al., 2007), these results suggest that the spatial bias in egg–sperm fusion is determined by RanGTP through F-actin–dependent and –independent pathways.

### Spatial separation of maternal and paternal chromosomes prevents the production of aneuploid zygotes

To test the importance of the spatial control of egg–sperm fusion sites, we directly placed paternal chromosomes in close proximity to maternal chromosomes through ICSI (Kimura and Yanagimachi, 1995). We placed paternal chromosomes within the 20-μm region surrounding maternal chromosomes and monitored the subsequent behavior of the paternal chromosomes (Fig. 6, A and B). In 20% of zygotes, the paternal chromosomes moved away from the maternal chromosomes along the plasma membrane (move away, 20%; Video 6). This observation

is consistent with the idea that zygotes actively keep paternal chromosomes away from maternal chromosomes after fertilization. However, in 28% of eggs, the signals of the paternal chromosomes were fused with those of the maternal chromosomes exhibiting anaphase motion and then eliminated into the polar body (polar body, 28%). In 34% of eggs, the signals of the paternal chromosomes were fused with those of the maternal chromosomes, which resulted in the formation of one diploid pronucleus in the zygote (fusion, 34%). When the paternal chromosomes were placed outside of the 20-μm region surrounding the maternal chromosomes, we did not observe the fusion of the signal of paternal chromosomes with those of maternal chromosomes or the elimination of paternal chromosomes into the polar body. These results suggest that the spatial control of egg–sperm fusion sites is critical to prevent the elimination of paternal chromosomes by maternal meiosis.

### F-actin keeps paternal chromosomes away from maternal chromosomes following egg–sperm fusion

Last, we investigated the mechanism that occurs after egg–sperm fusion regarding how paternal chromosomes remain at a distance from maternal chromosomes in the cytoplasm. We imaged F-actin using 3mEGFP_UtrCH during the fertilization process (Mori et al., 2011). After egg–sperm fusion, we observed three F-actin–rich cortical domains, that is, two actin caps surrounding each of the maternal anaphase chromosomes and a fertilization cone surrounding the paternal chromosomes, which were never fused to each other ($n$ = 31; Fig. 7 A and Video 7). F-actin inhibition with LatB induced the fusion of the signal of paternal chromosomes with those of maternal chromosomes when egg–sperm fusion occurred in proximity to maternal chromosomes (Fig. 7, B and C; and Video 8). To analyze the behavior of paternal chromosomes following egg–sperm fusion in proximity to maternal chromosomes, we placed sperm heads underneath the NA region (Fig. 7 D and Video 9). In control eggs, a substantial population of paternal chromosomes moved away from maternal chromosomes (11/22), while the signals of others were fused with those of maternal chromosomes. In contrast, in LatB-treated eggs, we never observed paternal chromosomes moving away from maternal chromosomes (0/16). These results suggest that an F-actin–dependent mechanism acts to separate paternal chromosomes away from maternal chromosomes following fertilization. This mechanism is required but not sufficient to guarantee the protection of paternal chromosomes in cases where egg–sperm fusion occurs in proximity to maternal chromosomes.

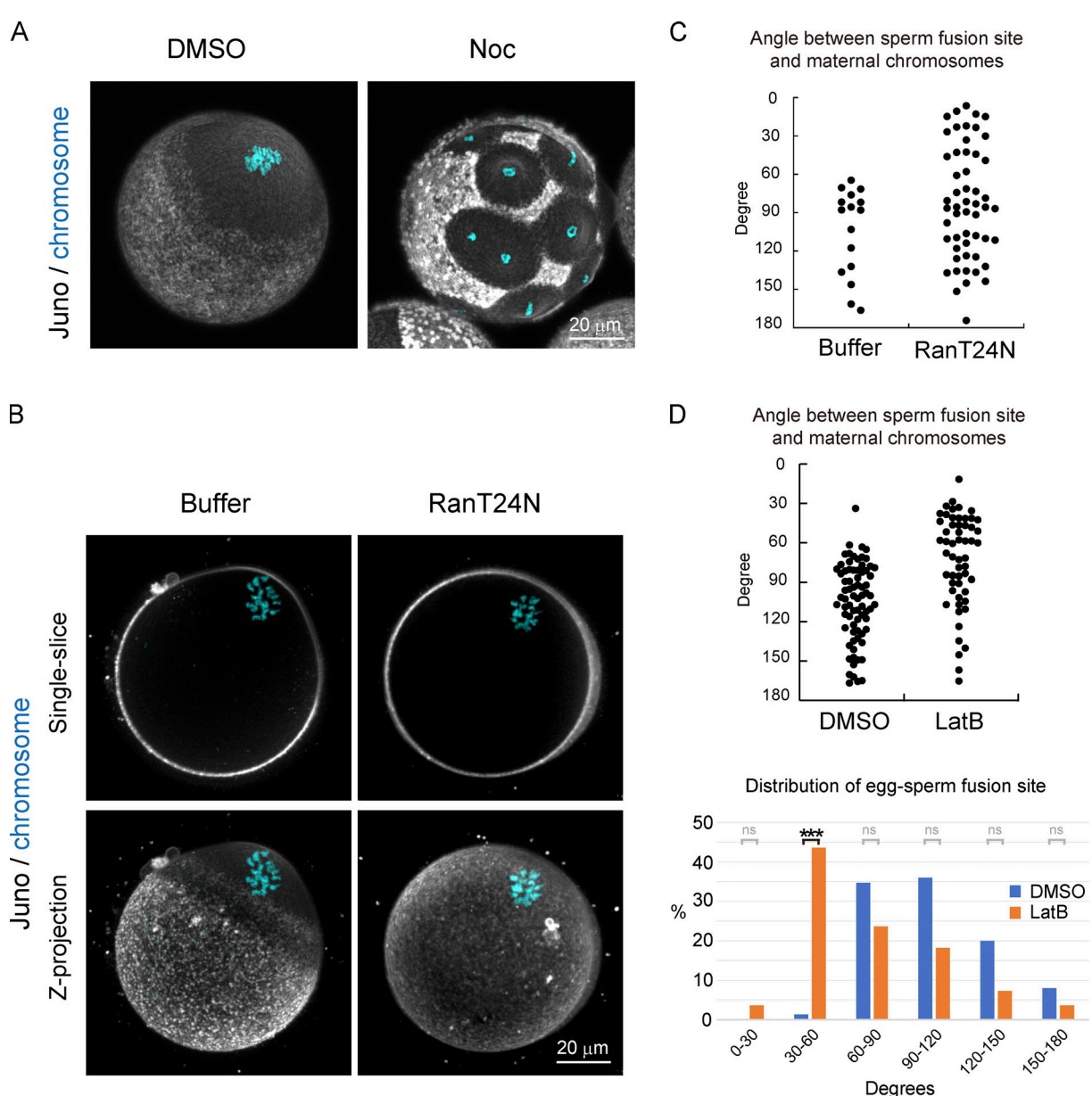

Figure 5. **RanGTP activity surrounding maternal chromosomes prevents the localization of Juno and CD9. (A)** Immunofluorescence images with fluorescent-labeled Juno and CD9 primary antibodies following treatment with DMSO or Noc (data of CD9 are not depicted). In Noc-treated eggs, Juno/CD9 structures are excluded around individual chromosomes. The zona pellucida was removed in A. **(B)** Immunofluorescence images with fluorescent-labeled Juno and CD9 primary antibodies in buffer and in RanT24N-injected eggs (data of CD9 is not depicted). In RanT24N-injected eggs, Juno/CD9 structures are formed all over the cell surface, even at regions proximal to maternal chromosomes. **(C)** Angle between maternal chromosomes and sperm fusion sites in buffer- or RanT24N-injected eggs. In RanT24N-injected eggs, sperm fuse at positions all over the egg surface. Data include zygotes that were polyspermic (polyspermy is 40% [buffer] and 48% [RanT24N] of the fertilized eggs). The zona pellucida was softened by treatment with glutathione. **(D)** Angle between maternal chromosomes and sperm fusion sites following treatment with DMSO or LatB. LatB-treated eggs exhibited significantly less spatial bias in sperm fusion compared with control eggs. However, sperm still did not fuse around maternal chromosomes. Data include zygotes that were polyspermic (polyspermy is 36% [DMSO] and 44% [LatB] of the fertilized eggs). A hole was made in the zona pellucida using a piezo-driven pipette without glutathione treatment. Fisher's exact tests were used to obtain P values, and Holm correction was used for the correction of multiple comparison (P value of 30–60° is <8.22 × 10$^{-10}$; P values of other areas are >0.05 [ns]). ***, P < 0.001.

## Discussion

The results of this study demonstrate that RanGTP signaling and F-actin dynamics play central roles in spatially separating paternal chromosomes from maternal chromosomes in zygotes (shown schematically in Fig. 8). Before fertilization, RanGTP signaling prevents the localization of Juno/CD9 structures on the cell surface in proximity to maternal chromosomes. Moreover, Juno/CD9 structures, together with F-actin–based lamellipodia-like protrusions, are transported to cell surface regions distal to maternal chromosomes. Accordingly, RanGTP activity and F-actin are required for biasing egg–sperm fusion sites toward regions distal to maternal chromosomes. Following egg–sperm

Figure 6. **Spatial separation of maternal and paternal chromosomes prevents the production of aneuploid zygotes. (A)** Live imaging of chromosomes (mCherry-MBD-NLS) in eggs in which sperm heads were injected within the 20-μm region surrounding maternal chromosomes. Four phenotypes were observed: move away, polar body, fusion, and center. In the center type, paternal chromosomes are not localized at the periphery of the zygote and move randomly in the cytoplasm. Time is relative to the appearance of the sperm signal. Time-lapse video of polar body and fusion are shown in Video 6. The zona pellucida was softened by treatment with glutathione (A and B). **(B)** Percentage of phenotypes in eggs in which sperm heads were injected outside or inside the 20-μm region surrounding maternal chromosomes. Right panel shows where sperm heads were injected. The values of the positions were normalized by the length between maternal chromosomes and the center of the egg when the paternal chromosome signal appeared.

fusion, F-actin keeps paternal chromosomes away from maternal chromosomes. These mechanisms enable the spatial separation of paternal chromosomes from maternal chromosomes, which is critical to prevent paternal chromosomes from being eliminated by maternal meiosis.

**RanGTP activity prevents the localization of Juno/CD9 structures on the cell surface in proximity to maternal chromosomes**

The initial position of paternal chromosomes upon fertilization depends on the sperm fusion site on the egg surface. The spatial

bias in egg–sperm fusion is primarily determined by maternal chromosome–mediated RanGTP activity, which prevents the localization of Juno/CD9 structures at the cell surface in proximity to maternal chromosomes. The de novo appearance of Juno/CD9 structures was observed at regions distal to maternal chromosomes but not in proximity to maternal chromosomes, which suggests that RanGTP activity inhibits the assembly and/ or recruitment of Juno/CD9 structures on the cell surface. Although the actin cap is a downstream target of RanGTP (Deng et al., 2007), the disruption of F-actin, including the actin cap, did not fully recapitulate the effects of RanGTP inhibition, which

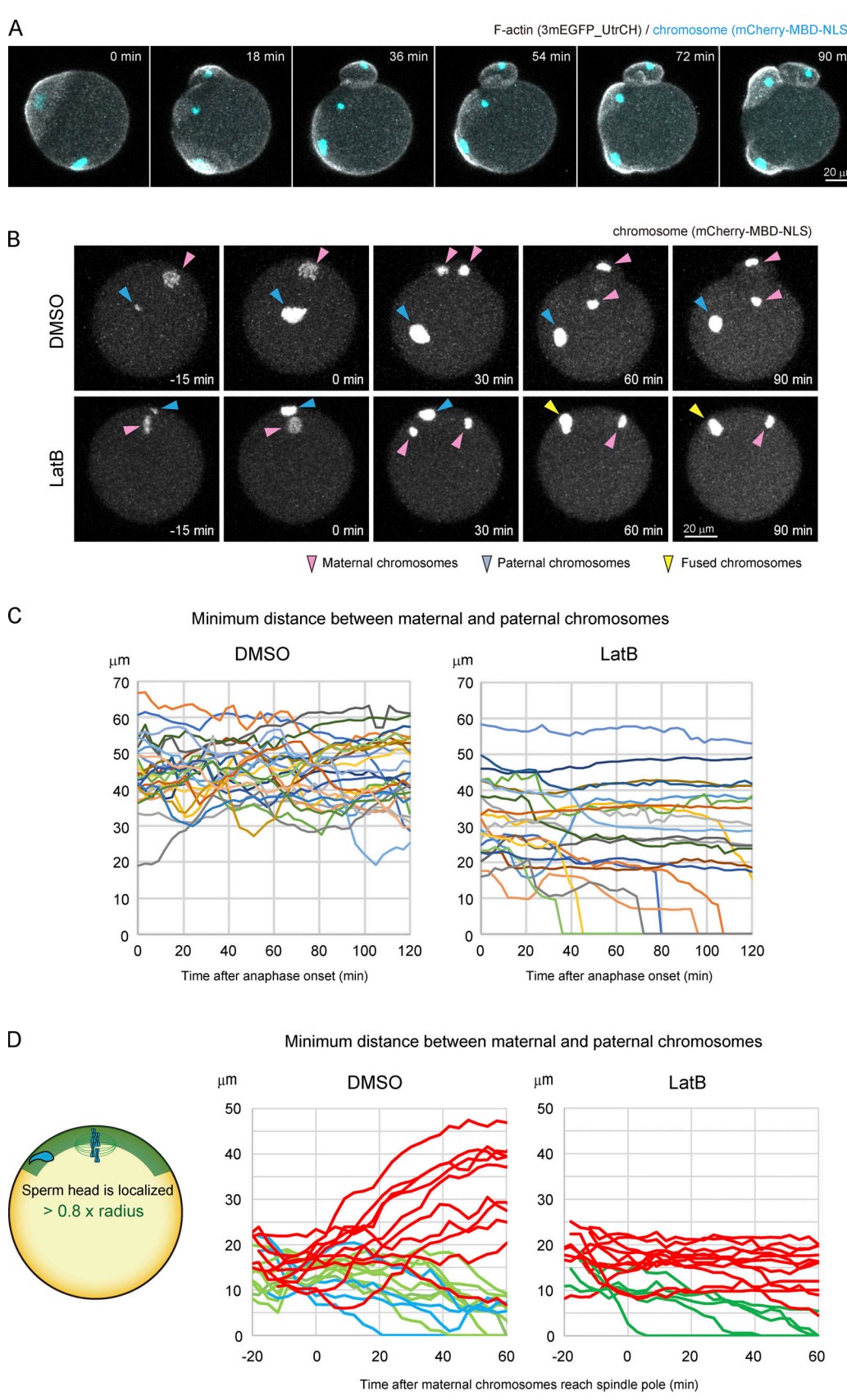

Figure 7. **F-actin keeps paternal chromosomes away from maternal chromosomes following egg–sperm fusion. (A)** Live imaging of F-actin and chromosomes using 3mEGFP_UtrCH and mCherry-MBD-NLS during the fertilization process. Maximum-intensity Z projection of F-actin and chromosome

signals are shown. Time is relative to anaphase onset. A time-lapse video is shown in Video 7. A hole was made in the zona pellucida using a piezo-driven pipette without glutathione treatment (A–C). **(B)** Live imaging of chromosomes (mCherry-MBD-NLS) following treatment with DMSO or LatB. Time is relative to anaphase onset. A time-lapse video is shown in Video 8. **(C)** Distance between paternal chromosomes and maternal chromosomes, which are closer after anaphase onset following treatment with DMSO or LatB. Paternal chromosomes are frequently captured by maternal chromosomes when a sperm fuses to a site within the 30-μm region surrounding maternal chromosomes. **(D)** Distance between paternal chromosomes and maternal chromosomes in eggs in which sperm heads were injected underneath the NA region following treatment with DMSO or LatB. A time-lapse video is shown in Video 9.

suggests that RanGTP activity can prevent the localization of Juno/CD9 structures in an F-actin–independent manner. RanGTP-dependent and F-actin–independent regulation of the egg surface, such as the dephosphorylation of the membrane protein Moesin (Dehapiot and Halet, 2013), may be involved in preventing the localization of Juno/CD9 structures. The disruption of F-actin significantly increased Juno/CD9 structures at regions proximal to maternal chromosomes, which suggests that F-actin is partially involved in the spatial control of Juno/CD9 structures. Furthermore, recently it was reported that CD9 functions to exclude Juno from the actin cap region (Inoue et al., 2020). RanGTP may regulate the localization of Juno through CD9 protein.

### A ring-like segment of the cell surface enriches lamellipodia-like protrusions and transports Juno/CD9 structures to regions distal to maternal chromosomes

This study identified a ring-like segment on the egg surface that is distinct from previously described segments such as the actin cap and the microvillus-covered region (Longo and Chen, 1985; Verlhac et al., 2000). The ring-like segment is enriched with Juno/CD9 structures in immunofluorescence images and with lamellipodia-like protrusions in EM images, which suggests that the Juno/CD9 structures are associated with lamellipodia-like

protrusions. We observed that Juno/CD9 structures in the FA region move toward the FA/DA border, but these structures in the DA region are relatively stable. Moreover, sperm bound in the FA region are transferred toward the DA region before their fusion on the egg surface. From these observations, the ring-like segment may act as a conveyor system that transports bound sperm to regions distal to maternal chromosomes. Since the ring-like segment is blocked by LatB, F-actin likely contributes to the transport of bound sperm, possibly by forming lamellipodia-like protrusions or by creating a cytoplasmic flow along the cell surface.

### F-actin keeps paternal chromosomes away from maternal chromosomes following egg–sperm fusion

After egg–sperm fusion, F-actin keeps paternal chromosomes away from maternal chromosomes, which is possibly mediated by cytoplasmic flow and cell surface regulation. It was already shown that cytoplasmic flow is generated by the actin cap, which enriches F-actin nucleators and regulators in mouse eggs (Yi et al., 2011). Moreover, the fertilization cone, an F-actin–rich cell surface domain containing thick F-actin bundles, forms surrounding paternal chromosomes. The fertilization cone may produce F-actin flow along the cell surface, as well as the actin cap surrounding maternal chromosomes, which may move

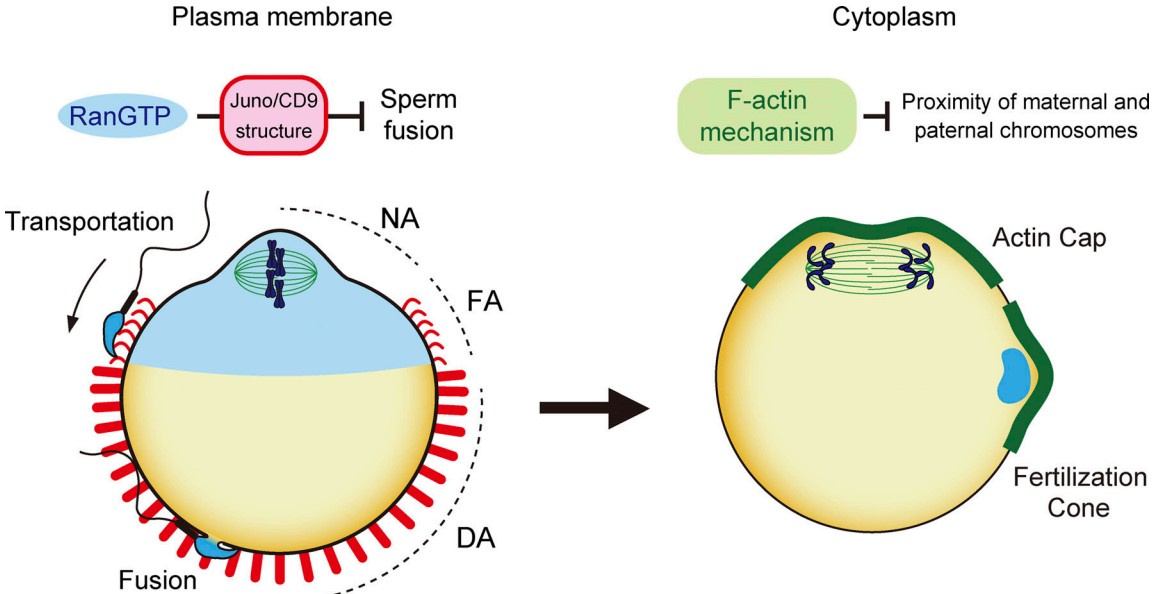

**Figure 8. Schematic of the mechanism by which zygotes regulate the localization of paternal chromosomes.** RanGTP signaling and F-actin dynamics play central roles in spatially separating paternal chromosomes from maternal chromosomes in zygotes. On the plasma membrane before fertilization, RanGTP regulates Juno/CD9 structures and blocks egg–sperm fusion in proximity to maternal chromosomes. In the cytoplasm, following egg–sperm fusion, F-actin keeps paternal chromosomes away from maternal chromosomes.

maternal and paternal chromosomes away from each other. Alternatively, the actin cap and the fertilization cone may act as independent semicompartments that separately accommodate maternal and paternal chromosomes, respectively. Consistent with this idea, live imaging showed that the actin cap and the fertilization cone were occasionally positioned next to each other but were never fused (Fig. 7, A and D; Video 7; and Video 9).

### Implications for human zygotes and assisted reproductive technologies

The spatial separation of paternal and maternal chromosomes is likely critical for human zygotes, which arrest at metaphase II before fertilization, as do mouse zygotes. In humans, 1.6–7.7% of embryos contain a single pronucleus in assisted reproductive technology, including ICSI. Embryos with a single pronucleus can result in viable pregnancies with no apparent anomalies, but their overall success rate of blastocyst formation is lower than that of embryos with two pronuclei (Itoi et al., 2015). According to our study, it is expected that embryos with a single pronucleus are formed because paternal chromosomes are discarded into the polar body or are captured by maternal chromosomes in embryos. The guidelines for ICSI, a key procedure for assisted reproductive technologies, recommend that the sperm injection site should be away from the polar body of the egg (European Society of Human Reproduction and Embryology [ESHRE], guidelines for good practice in IVF laboratories). However, an important caveat is that the polar body does not always mark the position of maternal chromosomes (Hewitson et al., 1999, Hardarson et al., 2000). We observed that, in mouse eggs, not only paternal but also maternal chromosomes moved along the cell surface, which frequently resulted in positioning the maternal spindle away from the polar body. Noninvasive visualization of the meiotic spindle would help prevent the production of aneuploid zygotes during ICSI procedures.

## Materials and methods

### Animal experimentation

All animal experiments were approved by the Animal Care and Use Committee of the Research Institute for Microbial Diseases, Osaka University (approval number 13003), and the Riken Center for Biosystems Dynamics Research (approval number A2011-05-15).

### Culture and microinjection of mouse eggs

Wild-type mice were purchased from CLEA Japan or Japan SLC. Female B6D2F1 mice (>8 wk old) were hyperovulated by injecting 5 U of human chorionic gonadotropin (hCG) 48 h after injecting 0.1 ml CARD HyperOva (Kyudo Company). Eggs were collected from the oviducts and placed in TYH medium (119.37 mM NaCl, 4.78 mM KCl, 1.19 mM $KH_2PO_4$, 5.56 mM glucose, 1 mM sodium pyruvate, 100 U/ml penicillin, 100 µg/ml streptomycin, 1.71 mM $CaCl_2$, 1.19 mM $MgSO_4$, 25.07 mM $NaHCO_3$, 4 mg/ml BSA [ALBMAX I, Gibco; or A3311, Sigma-Aldrich]), potassium-supplemented simplex optimized medium (KSOM), or M2 medium at 37°C and 5% $CO_2$. To

remove cumulus cells, eggs were treated with 1 mg/ml hyaluronidase (Sigma-Aldrich) for 5 min at 37°C. For Juno/CD9 immunofluorescence and EM experiments, the zona pellucida was removed by treatment with 1 mg/ml collagenase (Sigma-Aldrich) for 5 min. In vitro mRNA transcription was performed using a mMessage mMachine T7 Kit (Ambion). After microinjection of mCherry-MBD-NLS (1.2 pg) or 3mEGFP_UtrCH (1.2 pg) mRNA using a micromanipulator (Narishige) and a piezo-driven pipette (Prime Tech) on an inverted microscope (Olympus), eggs were cultured for 2 h before imaging. His-Ran was diluted to a final concentration of 1.5 µg/µl (in PBS containing 1 mM DTT and 0.05% Tween 20), and 50 pl was used for injection.

### Time-lapse imaging of in vitro fertilization

To soften the zona pellucida, eggs were treated with 0.5–1.0 mM glutathione in TYH medium for 30 min. Because the zona pellucida was expanded evenly and detached from the egg completely by the treatment, the penetration and fusion sites of sperm are likely random (fertilization rate, 58%; polyspermy rate, 43%). To reduce polyspermy, one or two holes were made in the zona pellucida using a piezo-driven pipette (Prime Tech) without glutathione treatment in the experiments shown in Fig. 3, D and F; Fig. 5 D; and Fig. 7, A and B. Because we made the hole beside the perivitelline space, which is often formed at the outer surface of the maternal spindle, the penetration site of sperm has a bias (fertilization rate, 46%; polyspermy rate, 24%). After injection of mCherry-MBD mRNA in M2 medium, these eggs were washed with BSA-free TYH medium several times and allowed to settle in a glass-bottom dish in a drop of BSA-free TYH medium. Before insemination, the same volume of TYH medium (with BSA) was added to the drop and incubated at 37°C. Epididymal sperm was collected from adult male mice (WT or transgenic mice, CAG/Su9-DsRed2, Acr3-EGFP) and incubated in TYH medium for 2 h for capacitation. On the microscope stage, capacitated sperm were added to the drop containing eggs at a final concentration of $10^5$ sperm/ml, and imaging was started immediately at 37°C. Images were typically acquired with a confocal microscope on a 100–120-µm square image plane, at a spatial resolution of 3 µm confocal sections covering 81–96 µm, and a temporal resolution of 2.5 or 3 min. We often recorded 10–30 eggs in parallel.

### Tracking of chromosomes and sperm tails

XY coordinates of paternal and maternal chromosomes, egg centers and sperm tails were obtained using a manual tracking plugin in ImageJ (National Institutes of Health) or the spot tracking function in Imaris (Bitplane). The angles and distances between paternal and maternal chromosomes were measured using these XY coordinates. For simulations to measure the angle and distance between the sperm fusion site and maternal chromosomes, we hypothesized that the egg is spherical and that sperm can fuse anywhere with equal probability. 40,000 random points on the sphere were generated using the runif_on_sphere function implemented in R package uniformly (Laurent, 2018), and the distance and angles were measured with the following parameters: the radius of the egg is 36.6 µm, and maternal chromosomes are localized

5 μm from the plasma membrane. The number was counted in each bin and corrected by the surface area. The angular displacement of paternal chromosomes toward maternal chromosomes was calculated between two time points ($n$ and $n + 2$), and the angle was divided by time to unify all data. To assess the relationship between the angular displacement of paternal chromosomes and their position along with time from the sperm fusion, multiple regression analysis was conducted. The predictor variable (the angle position of paternal chromosomes and the time after sperm fusion) from one to both and the interaction were considered in the linear regression model. The models were evaluated based on AIC values.

## ICSI

Epididymal sperm was mixed with HEPES-buffered CZB or M2 medium containing 12% (x/v) polyvinylpyrrolidone in a micromanipulation chamber. The head of each sperm was separated from the tail by applying pulses to the head–tail junction using a piezo-driven pipette (Prime Tech). The sperm head was injected into the spindle area of each egg, and imaging was started within 40 min after injection.

## Immunofluorescence

For Juno and CD9 staining, Alexa Fluor 647–conjugated anti-mouse Folate Receptor 4 (alternative name for Juno, 1/1,000), FITC-conjugated anti-mouse CD9 antibody (1/500; both from BioLegend, #125009 and #124807) and Hoechst 33342 (10 μg/ml, Invitrogen) were added into M2 medium and eggs with the zona pellucida (only in Fig. 5 B), or zona pellucida–free eggs were incubated in these media for 15 min. The viable eggs were washed with fresh medium and imaged. XY coordinates of Juno/CD9 structures and chromosomes were obtained manually using Imaris or ImageJ (Fig. 2 B). The angle between Juno/CD9 structures and chromosomes was measured using these XY coordinates, and angles of roughly 10 locations per egg were averaged. In Fig. 4 C, XY coordinates of DMSO_FA/DA and LatB_NA/DA were obtained as follows. We plotted the intensity of CD9 along a line on the maximum z-projection of the image, and the resultant plot was used to acquire the XY coordinates of DA border (local intensity increase). The XY coordinates were used to calculate the angle with the center of the egg in 3D. We acquired such angles from three lines per oocyte and averaged them. XY coordinates of DMSO_NA/FA and LatB_closest focus were obtained manually as in Fig. 2 B. For time-lapse imaging of Juno and CD9, images were typically acquired on a 50–100-μm square image plane, a spatial resolution of 2 μm confocal sections covering 60 μm, and a temporal resolution of 1–3 min with a scanning confocal microscope and a 40× lens. One z series takes ~15 s. For actin staining, eggs were fixed at room temperature for 30 min using fixation buffer (100 mM Pipes, pH 7.0, 1 mM MgCl₂, 0.1% Triton X-100, and 1.6% paraformaldehyde). After blocking with 3% BSA containing PBS for 30 min, the fixed eggs were incubated with Alexa Fluor 568 phalloidin (1/40, Life Technologies, #A12380) and Hoechst 33342 (10 μg/ml, Invitrogen) in PBS containing 3% BSA. These stained eggs were washed and imaged using a confocal microscope.

## EM

Solutions containing eggs were poured into a Mini cell strainer II (mesh size 40 μm; Hi-Tech) and preserved in the strainer throughout the procedure until they were mounted on double-sided carbon tape placed on a metal stub for EM observation. Otherwise, cells were absorbed on glass coverslips and processed for observation similarly as shown below. The samples were fixed with 2% paraformaldehyde and 2.5% glutaraldehyde in 0.1 M sodium cacodylate buffer, pH 7.4, for 2 h at room temperature. After washing with sodium cacodylate buffer three times, they were postfixed with 1% $OsO_4$ in sodium cacodylate buffer for 1 h on ice. The samples were then washed with water and dehydrated with graded concentrations of ethanol. After substitution with $t$-butyl alcohol, they were dried using a freeze-drying device (JEOL JFD-320). After mounting, samples were coated with osmium using an Osmium coater (Neoc-Pro/P, Meiwafosis) and observed using a scanning electron microscope (JEOL JSM 5600 LV).

## Purification of Ran recombinant protein

Ran cDNA was amplified by PCR using primers (5′-AATCATATG GCCGCCCAGGGAGAGC-3′ and 5′-AATGAATTCTCACAGGTC ATCATCCTCATCTGGGAGAG-3′) and cloned into the pET28b vector using NdeI and EcoRI. The T24N mutant was generated using a KOD Plus Mutagenesis Kit (Toyobo) with primers (5′-GCACCGGGAAGAATACCTTCGTGAAGCGCCACTTGACGGGCG AG-3′ and 5′-GCTTCACGAAGGTATTCTTCCCGGTGCCGCCGT CGCCCACCAGG-3′). pET28b-RanWT or pET28b-RanT24N plasmids were transformed into BL21(DE3). His-Ran recombinant proteins were expressed with 1 mM IPTG (Nacalai) at 28°C for 2 h and purified by the Qiagen Ni-NTA protocol for native conditions (Qiagen; lysis buffer: 50 mM $NaH_2PO_4$, 300 mM NaCl, 10 mM imidazole, pH 8.0; wash buffer: 50 mM $NaH_2PO_4$, 300 mM NaCl, 20 mM imidazole, pH 8.0; and elution buffer: 50 mM $NaH_2PO_4$, 300 mM NaCl, 250 mM imidazole, pH 8.0). The buffer of the eluted fraction was changed to PBS using a PD-10 column (GE), and His-Ran proteins were concentrated using a Vivaspin 500 (GE).

## Microscopy

Live imaging was performed using a spinning disk confocal microscope (Olympus microscope with CSU-X1 [Yokogawa Electric] and an electron-multiplying charge-coupled device camera [Andor]) with Volocity software or a scanning confocal microscope (Zeiss 710, 780, 880) with Zen software. Depending on the experiment, we chose an appropriate lens from an oil-immersion 20× lens, a silicone-immersion 30× or 60× lens, a water-immersion 60× lens (Olympus), a water-immersion 40× lens, and an oil-immersion 100× lens (Zeiss). All images shown were processed by a Gaussian filter in ImageJ to remove detector noise.

## Statistical analysis

All statistical analyses were performed using R, Prism 9 (GraphPad Software), or Excel (Microsoft). Fisher's exact tests were used for Figs. 1 B, S1 B, and S5 C using expected assignment probability to each bin resulting from the simulation as a null

hypothesis. Holm correction was used for the correction of multiple comparisons. Fisher's exact test was used for Fig. 5 D. Welch's *t* test was used for Figs. 1 F, 4 C, and S1 D. All analyses were two-sided tests. P values <0.05 are considered statistically significant: *, P < 0.05; **, P < 0.01; and ***, P < 0.001 in the figures.

## Measurement of the mechanical properties of the zona pellucida

The mechanical properties of the zona pellucida were examined using a dual microneedle–based setup, which we previously developed to analyze the mechanical properties of human cell nuclei (Shimamoto et al., 2017). The setup was built in an inverted microscope (Ti-U, Nikon), which was equipped a motorized X-Y stage (MS-2000, Applied Scientific Instruments) and a pair of three-axis hydraulic micromanipulators (MHW-3, Narishige), with which microneedles were held and stirred. A 40× objective lens (Plan Apo, 0.95 NA; Nikon), a scientific complementary metal–oxide–semiconductor camera (Neo4.1, Andor), and imaging software (NIS elements v5.0, Nikon) were used to acquire images. The microneedles were prepared by microfabricating glass rods (G-1000, Narishige) using a capillary puller (PD-10; Narishige) and a microforge (MF-900, World Precision Instruments) so that their tips were a uniform, fiber-like shape with a diameter of ~1 μm (Shimamoto and Kapoor, 2012).

The mechanical measurement was performed using the microneedles as prepared above whose tips had different values of stiffness. Specifically, the tip of one microneedle was significantly stiff, such that it did not bend during the mechanical actuation. The tip of the other microneedle was much more flexible and thus could bend in response to the force that it pushed. This microneedle could act as a force sensor, as the displacement of the tip was proportional to the amount of force applied. The tips of the two microneedles were brought into contact with the surface of the zona pellucida from opposite sides of the egg. Subsequently, the stiff microneedle was moved closer to the flexible microneedle to apply compression, and fixed at the new position until the force reached a steady level, and finally returned to the original position so that the applied force could be removed. This procedure was repeated with varying compression magnitudes such that the stiffness of the zona pellucida could be examined over a range of deformation forces. After completing the measurements at a point of interest (e.g., the point proximal to the egg's protruding cortex), the egg was rotated by 90° horizontal to the imaging plane using the two microneedles, and the next targeting point was attached to the tip of the flexible microneedle. Unfertilized eggs, which were prepared freshly on the same day of the experiment, were maintained in KSOM medium (ARK Resource) at 37°C and 5% $CO_2$ in a cell culture incubator (Forma, Thermo Fisher Scientific) before being transferred into film-bottom dishes (FD20301, Matsunami) with the medium and then covered with paraffin oil (26137–85, Nacalai Tesque). Measurements were performed at 37°C in a stage-top incubator (INU-TIZ; Tokai Hit) and completed within 60 min after transferring eggs into the dishes,

over which time no recognizable differences in the mechanics or morphology of the eggs were found.

The local stiffness of the zona pellucida was determined based on the force–deformation relationship obtained from the measurements. The amount of force ($F$) could be calculated by multiplying the displacement of the force-sensing tip of the microneedle from its equilibrium position ($\Delta X$) with its pre-calibrated stiffness ($kf$: 12.2 nN/μm), according to $F = kf \Delta X$. The extent of deformation ($\Delta D$) was measured by analyzing the position change of the outer surface of the zona pellucida at the contact point of the flexible microneedle. Repeating the measurements with different force magnitudes allowed for generation of the force–deformation relationship, whose slope as determined by linear regression yielded the local stiffness of the zona pellucida. The analyses were performed using ImageJ (v1.48v) and Excel. Statistical analysis was performed in Prism v9.1.0.

## Measurement of the porosity of the zona pellucida

The porosity of the zona pellucida was analyzed by soaking unfertilized eggs in KSOM medium containing 0.13 mg/ml TRITC-dextran (T1162, Sigma-Aldrich). For the measurement, the eggs were maintained at 37°C and covered with paraffin oil. Confocal fluorescence images were acquired using the microscopy setup as described above with a spinning-disk confocal unit (CSU-X1, Yokogawa) and a single-mode excitation laser (OBIS561, Coherent). The fluorescence signal profile was obtained along the circumference of the zona pellucida by performing a circular intensity scan at the middle between its outer and inner surfaces (dotted line, Fig. S4 A). The scan width was set at 1.5 μm. The analysis was performed using ImageJ.

### Online supplemental material

Figs. S1 and S2 provide details of the tracking analysis of chromosomes during the fertilization process. Fig. S3 shows measurement of the mechanical properties of the zona pellucida. Fig. S4 shows measurement of the porosity of the zona pellucida. Fig. S5 presents actin structures in LatB-treated eggs and Ran-inhibited eggs. Video 1 shows sperm tail and paternal chromosomes. Video 2 shows maternal and paternal chromosomes during the fertilization process. Video 3 shows high-resolution imaging of Juno structures in FA and DA. Video 4 shows sperm tail and paternal chromosomes before sperm fusion. Video 5 shows Juno structures after DMSO or LatB treatment. Video 6 shows chromosome movements after ICSI. Video 7 shows F-actin and chromosomes during the fertilization. Video 8 shows chromosomes during the fertilization after DMSO or LatB treatment, and Video 9 shows chromosome movements after ICSI following DMSO or LatB treatment.

## Acknowledgments

We are thankful to members of the Ikawa group and the Kitajima group for reagents and support. We thank the Division for Development of Genetic-engineered Mouse Resource at the National Institute of Genetics for support. We thank Jan Ellenberg (EMBL) for sharing the pGEMHE_3mEGFP_UtrCH construct.

This work was supported by Grants-in-Aid for Scientific Research (KAKENHI) awards JP19H05750 (to M. Ikawa), JP17H05005 (to M. Mori), JP18H05528 (to K. Yamagata), JP25712035 (to K. Yamagata), JP18H02357 (to K. Yamagata), JP18H02617 (S. Yonemura), JP18H05549 (to T.S. Kitajima), JP19H05751 (to Y. Shimamoto), and JP21K14919 (to T. Mishina); Japan Agency for Medical Research and Development grant JP20gm5010001 (to M. Ikawa); the Naito Foundation (to M. Mori); Japan Society for the Promotion of Science grant 18J00928 (to T. Mishina); and the Takeda Science Foundation (to Y. Shimamoto). This research was funded, in part, by the Bill and Melinda Gates Foundation, grant INV-001902.

The authors declare no competing financial interests.

Author contributions: M. Ikawa, T. Yao, K. Yamagata, and M. Mori established the live imaging setting and started this project. T. Yao, K. Yamagata, N. Yonezawa, and M. Mori performed the experiments using microscopy. H. Endoh, M. Mori, and S. Yonemura performed the experiments using electron microscopy. T. Mishina performed the simulation of the sperm binding site, the evaluation of angular displacement models, and statistical analysis. M. Tanaka and Y. Shimamoto performed the measurements of mechanical properties and porosity of the zona pellucida. M. Ikawa, T.S. Kitajima, and M. Mori contributed to the design and interpretation of the experiments and to writing the manuscript.

Submitted: 1 December 2020

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

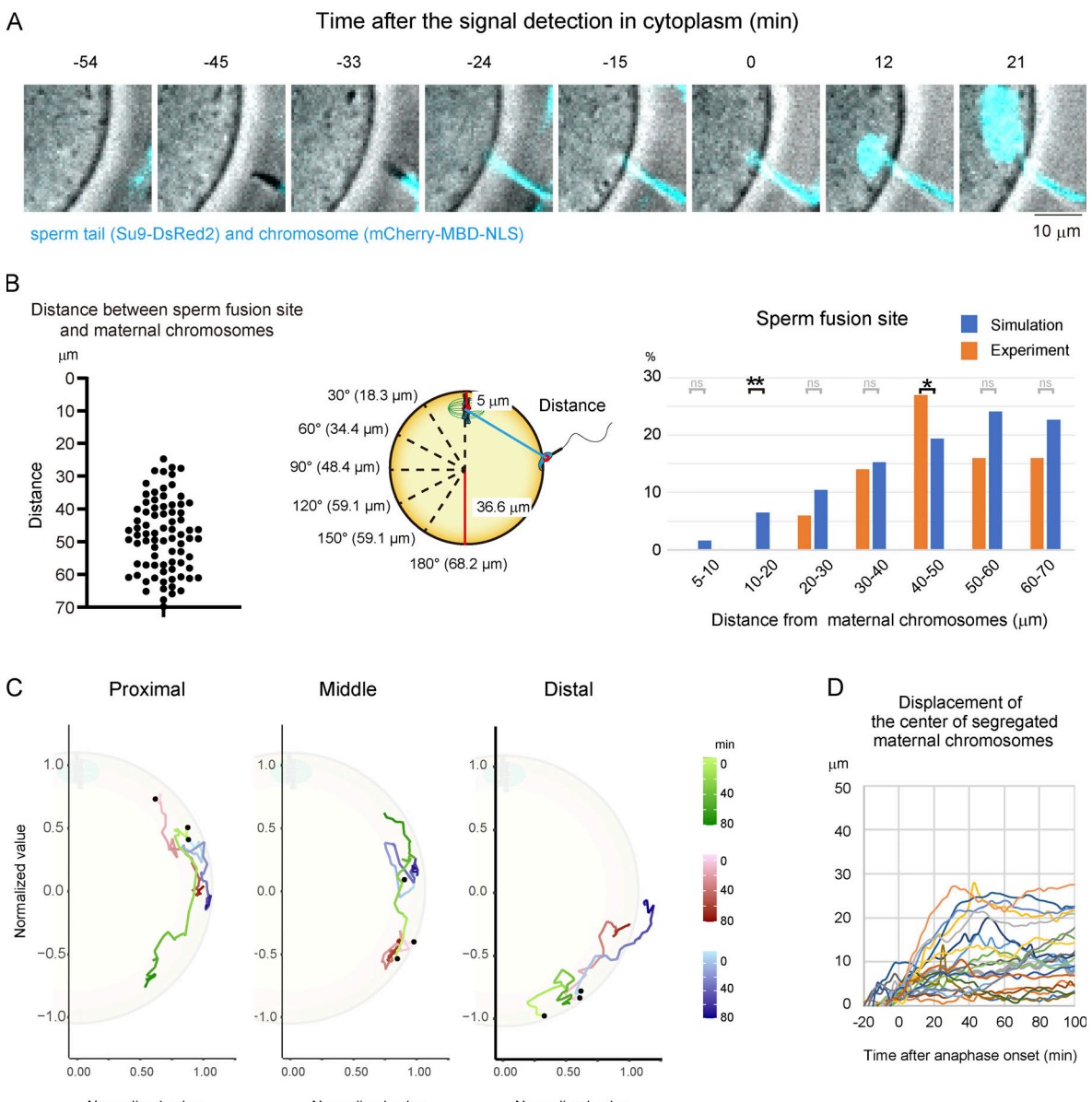

Figure S1. **Tracking analysis of chromosomes during the fertilization process. (A)** MBD showing where sperm fuse. The mCherry-MBD-NLS signal appears where the sperm tail (Su9-DsRed2) is bound. Time is relative to the sperm binding to the egg. A time-lapse video is shown in Video 1. The zona pellucida was softened by treatment with glutathione (A–D). **(B)** Distance between maternal chromosomes and sperm fusion sites. Left: Experimental data. Middle: Parameters for the simulation in which sperm fuse randomly on the egg surface. Right; Comparison between experimental and simulation data. Fisher's exact tests were used to obtain P values, and Holm correction was used for the correction of multiple comparisons (P value of 10–20 μm is $8.3 \times 10^{-3}$; P value of 40–50 μm is $4.0 \times 10^{-2}$; P values of other areas are >0.05 [ns]). **(C)** Examples of trajectory in Fig. 1 C are shown in three areas (proximal, middle, and distal from maternal chromosomes). Black dots show the position of sperm fusion. The values of the position of chromosomes were normalized by the length between the sperm fusion site and the center of the egg. **(D)** Displacement of maternal chromosomes from the initial position when sperm fuse. *, P < 0.05; **, P < 0.01.

## A
### Angle Only model

Angular displacement is correlated with the position of paternal chromosomes (Angle)

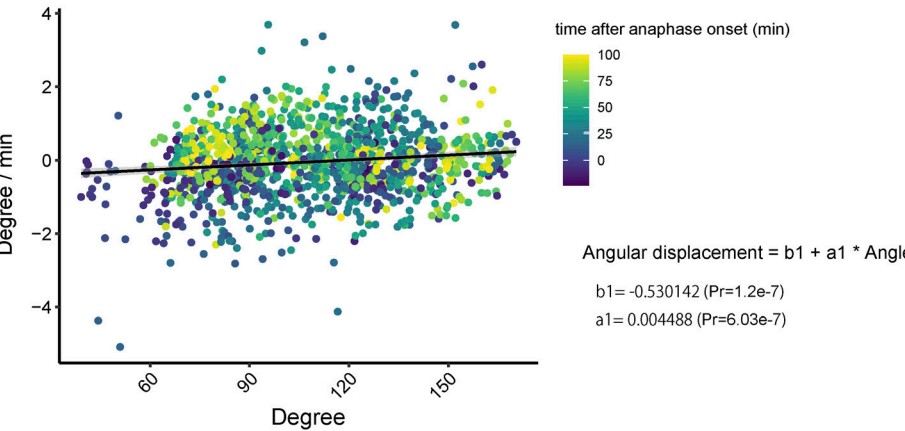

Angular displacement = b1 + a1 * Angle,

b1= -0.530142 (Pr=1.2e-7)
a1= 0.004488 (Pr=6.03e-7)

### Time Only model

Angular displacement is correlated with the Time after sperm fusion (Time)

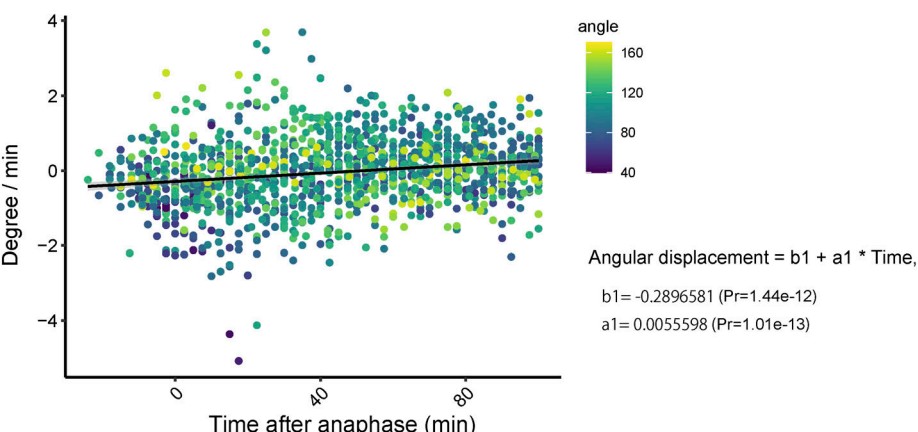

Angular displacement = b1 + a1 * Time,

b1= -0.2896581 (Pr=1.44e-12)
a1= 0.0055598 (Pr=1.01e-13)

## B
Maximum displacement of
paternal and maternal chromosomes
within 80 min after anaphase onset

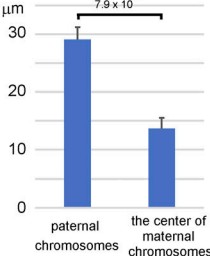

Figure S2. **Tracking analysis of chromosomes during the fertilization process. (A)** Angular displacement of paternal chromosomes toward maternal chromosomes versus the position of paternal chromosomes (angle only, top) and the time after sperm fusion (time only, bottom). **(B)** Average of the dataset in Figs. 1 D and S1 C. Welch's *t* test was used to obtain P value.

**Figure S3. Mechanical uniformity of the zona pellucida of unfertilized mouse eggs. (A and B)** Schematic illustration showing the microneedle-based setup used to examine the local mechanical properties of the zona pellucida. The setup consists of two glass microneedles whose motion can be controlled using hydraulic micromanipulators such that unfertilized mouse eggs surrounded by the zona pellucida are captured and subjected to a localized force (A). The force can be applied by moving the actuator microneedle in the indicated direction (black arrow, B). The amount of force can be monitored based on the deflection of the force-sensing tip of the microneedle from its equilibrium point (ΔX). The extent of deformation (ΔD), which predominantly arises around the area to which the tips of the microneedles are attached, can be measured by analyzing the local shape change of the zona pellucida (B). **(C–J)** Typical bright-field images (C–F) and force–deformation relationships (G–J) obtained from the measurements. The local stiffness of the zona pellucida was measured by attaching the force-sensing tip of the microneedle at the following four cardinal points: the points most proximal and distal to the egg's protruding cortex (white arrow; defined as 0 and 180°, respectively) and the points that were orthogonal to the 0–180° axis (defined as 90 and 270°). At each point of interest, the zona pellucida was deformed by applying cycles of compressive force with varying magnitudes, from which the force–deformation relationship was obtained (G–J). The egg was then rotated by 90° horizontal to the imaging plane to proceed in measuring the next cardinal point. The egg cell started to deform when the zona pellucida was compressed to a large extent (vertical arrows). However, the force–deformation relationship did not change significantly above that point, indicating that the contribution of the egg cell to the measured mechanics was minor. A total of $n = 4$ samples were examined (plotted in different colors in G–J). White solid and dotted lines in C–F highlight outlines of the zona pellucida before (upper panels) and during (lower panels) compression, respectively. At each compression, the deformation developed on both sides of the zona pellucida; the difference in the extent of deformation was 2.2 ± 6.3% (mean ± SD, $n = 201$ trials from 4 samples). Scale bars, 50 µm. **(K)** The local stiffness of the zona pellucida measured at the indicated cardinal points relative to the protruding cortex (0°). The stiffness value at each point was determined by performing linear regression in G–J for individual samples and was then averaged (2.03 ± 0.05, 2.05 ± 0.09, 2.05 ± 0.07, and 2.05 ± 0.15 nN/µm; mean ± SD, $n = 4$). ns, $P > 0.37$ by Student's $t$ test.

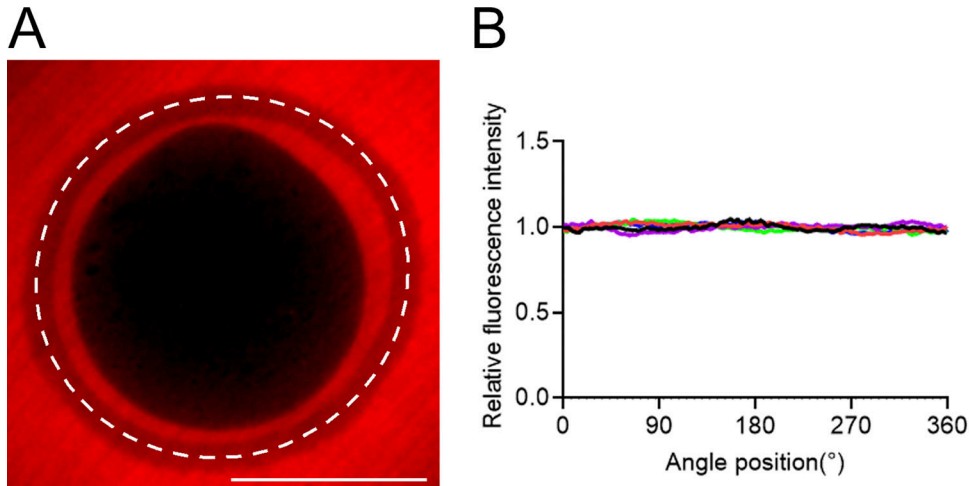

Figure S4. **Uniformity of the porosity of the zona pellucida of unfertilized mouse eggs. (A)** Unfertilized mouse eggs surrounded by the zona pellucida were soaked in KSOM medium containing 70 kD TRITC-dextran and imaged using confocal fluorescence microscopy. Scale bar, 50 μm. **(B)** An intensity scan was performed along the circumference of the zona pellucida (dotted line) to examine the uniformity of the structure. The 70-kD dextran has a hydrodynamic radius of ~6.5 nm (Armstrong et al., 2004) and thus could penetrate through the zona pellucida, filling the space within its porous structure. The fluorescence signal at the zona pellucida was higher than that within the egg cell, whose membrane is not permeable to dextran, and it was lower than in the surrounding buffer due to the porosity. Colored plots in B indicate data from different samples ($n = 5$). The angle 0° is defined as the point of the protruding cortex of the egg. The relative fluorescence intensity, as normalized based on the mean intensity of each circular scan, varied by 8.6 ± 0.7% ($n = 5$) along the circumference of the zona pellucida, but the intensity minima and maxima did not appear at any specific location.

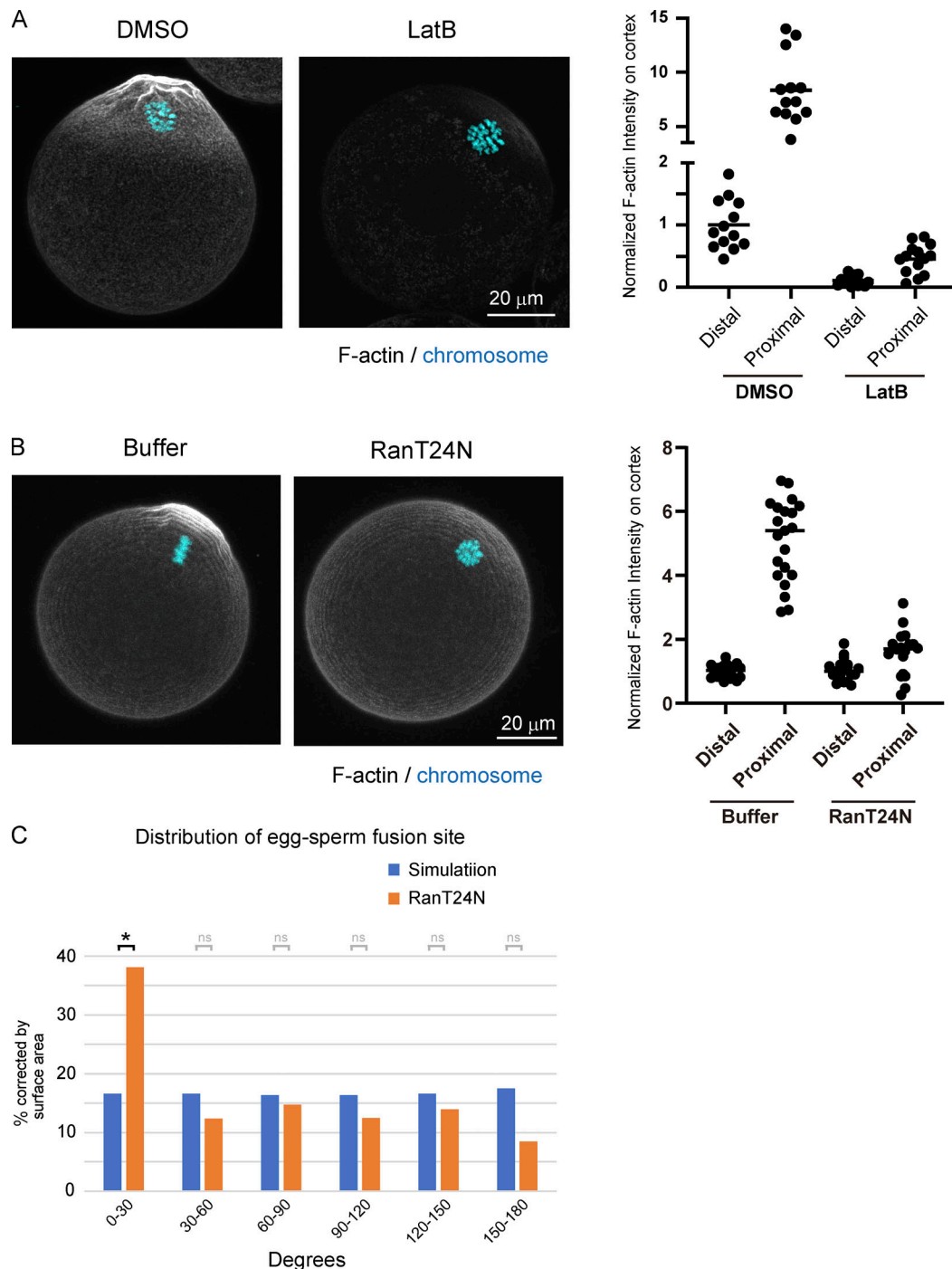

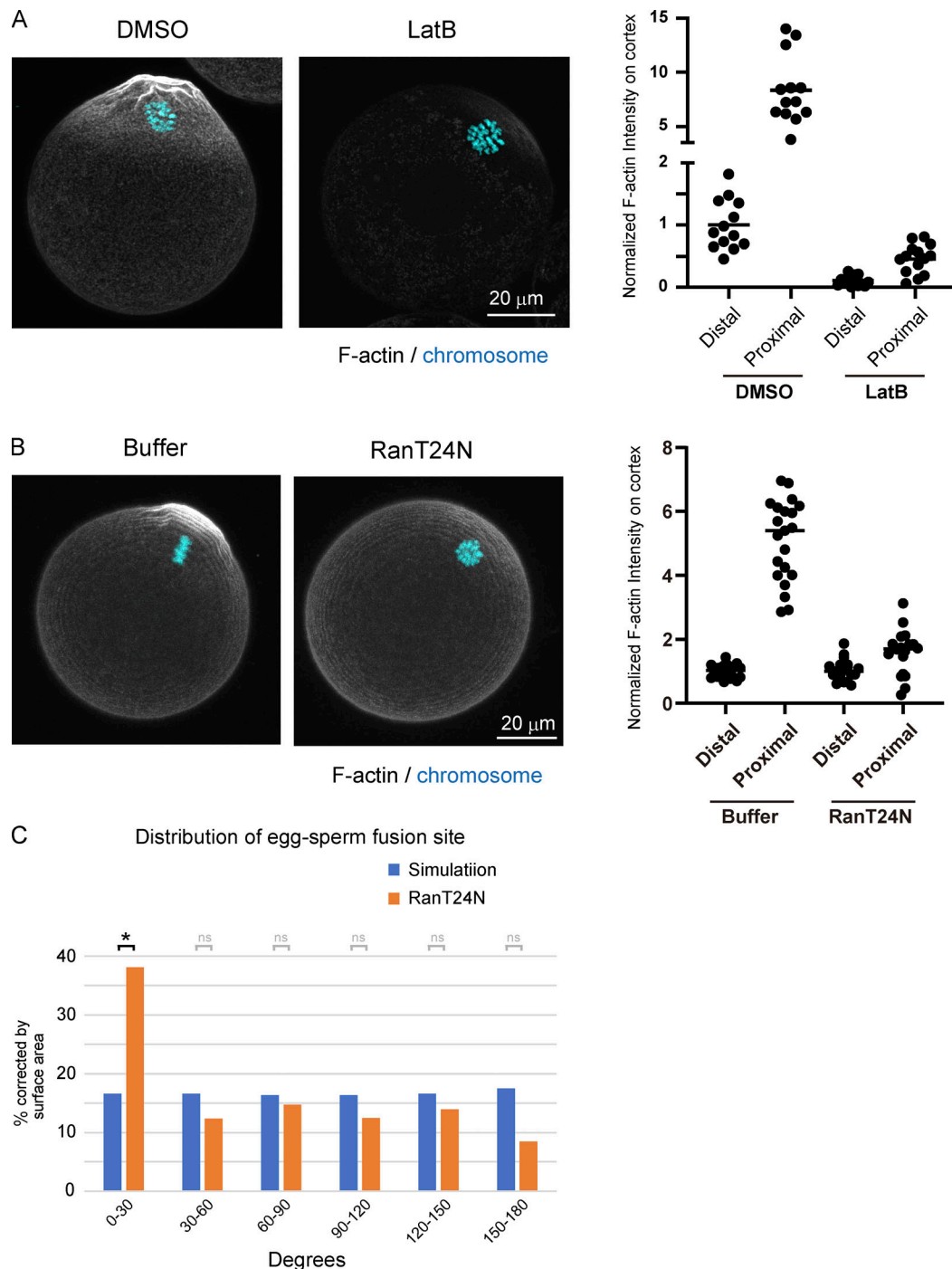

Figure S5.    **Actin structures in unfertilized eggs after treatment with LatB. (A)** Fluorescent staining of F-actin and chromosomes following treatment with DMSO or LatB. Eggs were fixed and stained with phalloidin and Hoechst 33342. The actin intensity was measured at the cortex near (proximal) or far (distal) from maternal chromosomes in eggs treated with DMSO or LatB. **(B)** Fluorescent staining of F-actin and chromosomes in Ran-inhibited eggs (RanT24N). The actin intensity was measured at the cortex near (proximal) or far (distal) from maternal chromosomes in Ran-inhibited eggs (RanT24N). **(C)** Angle between maternal chromosomes and sperm fusion sites in RanT24N-injected eggs. We compare the experimental data with the result of a simulation where sperm fuse randomly anywhere on the egg surface. Data include zygotes that exhibited polyspermy (polyspermy rate was 48% [RanT24N] of the fertilized eggs). Fisher's exact tests were used to obtain P values, and Holm correction was used for the correction of multiple comparisons (P value of 0–30° is < 3.7 × 10$^{-2}$; P values of other areas are >0.05 [ns]). *, P < 0.05.

Video 1.   **Live imaging of chromosomes (mCherry-MBD-NLS) and a sperm tail (Su9-DsRed2) during the fertilization process.** Time is relative to the sperm binding to the egg. The frame rate is 3 min per frame.

Video 2.   **Live imaging of chromosomes in an egg expressing mCherry-MBD-NLS during the fertilization process.** Time is relative to anaphase onset. The frame rate is 3 min per frame.

Video 3.   **Live imaging of Juno/CD9 structures with fluorescent-labeled Juno and CD9 primary antibodies using an Airyscan detector (data of CD9 are not depicted).** Time is relative to the start of imaging. The frame rate is 1 min per frame.

Video 4.   **Live imaging of chromosomes (mCherry-MBD-NLS) and sperm tail (Su9-DsRed2) during the fertilization process.** Two sperm bind and fuse with an egg. The sperm which bind in the FA region move away from maternal chromosomes. The frame rate is 3 min per frame.

Video 5.   **Live imaging of Juno structures with a fluorescent-labeled Juno primary antibody in DMSO- or LatB-treated eggs.** Since the images were taken using different microscopes in DMSO- or LatB-treated eggs, the intensities of Juno structures cannot be compared in these eggs. Time is relative to the addition of DMSO or LatB. The frame rate is 3 min per frame.

Video 6.   **Live imaging of chromosomes (mCherry-MBD-NLS) in eggs in which sperm heads were injected within the 20-µm region surrounding maternal chromosomes.** The four types of chromosome behavior are shown. Time is relative to anaphase onset. The frame rate is 2.5 min per frame.

Video 7.   **Live imaging of F-actin and chromosomes using 3mEGFP_UtrCH and mCherry-MBD-NLS during the fertilization process.** Time is relative to the start of imaging. The frame rate is 3 min per frame.

Video 8.   **Live imaging of chromosomes (mCherry-MBD-NLS) following treatment with DMSO or LatB during the fertilization process.** Since the images were taken using different microscopes in DMSO or LatB-treated eggs, the intensities of chromosomes cannot be compared in these eggs. Time is relative to anaphase onset. The frame rate is 3 min per frame.

Video 9.   **Live imaging of chromosomes (mCherry-MBD-NLS) in eggs in which sperm heads were injected underneath the NA region following treatment with DMSO or LatB.** Since the images were taken using different microscopes in DMSO or LatB-treated eggs, the intensities of chromosomes cannot be compared in these eggs. Time is relative to the time that maternal chromosomes reach spindle pole. The frame rate is 3 min per frame.

