## [Peer Review File · The Journal of Cell Biology]

RanGTP and the actin cytoskeleton keep paternal and maternal chromosomes apart during fertilization

Masashi Mori, Tatsuma Yao, Tappei Mishina, Hiromi Endoh, Masahito Tanaka, Nao Yonezawa, Yuta Shimamoto, Shigenobu Yonemura, Kazuo Yamagata, Tomoya Kitajima, and Masahito Ikawa

Corresponding Author(s): Masashi Mori, RIKEN Center for Developmental Biology and Masahito Ikawa, Osaka University

Review Timeline:

Submission Date:	2020-12-01
Editorial Decision:	2021-01-14
Revision Received:	2021-06-18
Editorial Decision:	2021-07-13
Revision Received:	2021-07-21

Monitoring Editor: Arshad Desai

Scientific Editor: Dan Simon

Transaction Report:

DOI: <https://doi.org/10.1083/jcb.202012001>

January 14, 2021

Re: JCB manuscript #202012001

Dr. Masashi Mori
RIKEN Center for Developmental Biology
2-2-3 Minatojima-minamimachi, Chuou-ku,
Kobe, Hyogo 650-0047
Japan

Dear Dr. Mori,

Thank you for submitting your manuscript entitled "RanGTP and F-actin prevent sperm fusion near the maternal spindle in mouse eggs". The manuscript was assessed by expert reviewers, whose comments are appended to this letter.

Overall, the reviewers are enthusiastic about your study stating that it addresses an important question and they are complimentary of the quality of the data presented in the manuscript. We invite you to submit a revision if you can address the reviewers' key concerns, as outlined here.

You will see that Reviewer #3 is concerned that the statistical analysis provided is not sufficient to conclude that fusion sites and paternal DNA movements in the zygote are actively directed away from the maternal DNA or if they are random. We agree that this is an important question that must be addressed during revision. Reviewer #2 raises specific points that we believe are addressable through clarification in the text and figures. Reviewer #1 is the most critical, indicating that while the quality of the data is high the work is largely descriptive; this reviewer suggests a number of additional experiments to extend the study. Given the support from the other reviewers, we do not believe significant additional experimental extension is necessary during revision but we encourage you to consider the points raised by Reviewer #1 and assess whether addition of existing data and/or text may aid in reducing the reviewer's major concern.

When submitting your revision, please include a detailed point-by-point response to the reviewers' comments indicating the changes made in the revised manuscript.

GENERAL GUIDELINES:

Text limits: Character count for an Article is < 40,000, not including spaces. Count includes title page, abstract, introduction, results, discussion, acknowledgments, and figure legends. Count does not include materials and methods, references, tables, or supplemental legends.

Figures: Articles may have up to 10 main text figures. Figures must be prepared according to the policies outlined in our Instructions to Authors, under Data Presentation, <https://jcb.rupress.org/site/misc/ifora.xhtml>. All figures in accepted manuscripts will be screened prior

to publication.

IMPORTANT: It is JCB policy that if requested, original data images must be made available. Failure to provide original images upon request will result in unavoidable delays in publication. Please ensure that you have access to all original microscopy and blot data images before submitting your revision.

Supplemental information: There are strict limits on the allowable amount of supplemental data. Articles may have up to 5 supplemental figures. Up to 10 supplemental videos or flash animations are allowed. A summary of all supplemental material should appear at the end of the Materials and methods section.

As you may know, the typical timeframe for revisions is three to four months. However, we at JCB realize that the implementation of social distancing and shelter in place measures that limit spread of COVID-19 also pose challenges to scientific researchers. Lab closures especially are preventing scientists from conducting experiments to further their research. Therefore, JCB has waived the revision time limit. We recommend that you reach out to the editors once your lab has reopened to decide on an appropriate time frame for resubmission. Please note that papers are generally considered through only one revision cycle, so any revised manuscript will likely be either accepted or rejected.

Thank you for this interesting contribution to Journal of Cell Biology. You can contact us at the journal office with any questions, cellbio@rockefeller.edu or call (212) 327-8588.

Sincerely,

Arshad Desai, Ph.D.
Monitoring Editor
Journal of Cell Biology

Dan Simon, Ph.D.
Scientific Editor
Journal of Cell Biology

Reviewer #1 (Comments to the Authors (Required)):

Authors are addressing here an important question: how is the haploid set of paternal chromosomes not extruded together with the maternal set of separating chromosomes following fertilization of mouse metaphase II oocytes? This is a long-standing question that have so far been partly resolved by prior works showing:

(1) that maternal chromosomes induce a cortical differentiation, which corresponds to a region

above them devoid of microvilli and enriched in actin, that does not favour sperm entry in proximity to maternal chromosomes (Maro J Emb Exp Morph 1984; Longo and Chen Dev Biol 1985). Similarly the paternal set of chromosomes also induces what has been referred as a fertilization cone, enriched in F-actin and devoid of microvilli (Shalgi Gamete Res 1978; Maro J Emb Exp Morph 1984; Longo and Chen Dev Biol 1985).

(2) that maternal chromosomes produce a Ran-GTP gradient potentially perturbed by the expression of Ran mutant forms (Dumont JCB 2007) that induces the above-mentioned cortical differentiation (Deng Dev Cell 2007) as well as a cytoplasmic flow that will tend to send sperm chromosomes away from maternal ones (Yi Nat Cell Biol 2011).

The novelty that authors are bringing here is that this cortical differentiated region is also devoid of two major players in sperm/egg membrane fusion, Juno and CD9. They show using beautiful and convincing live imaging that the localisation of these two proteins mimics that of microvilli, accumulating mostly where the protrusions are denser in the cortex of the oocyte. Using a series of perturbation experiments, authors also demonstrate that Juno and CD9 localisations depend on the presence of chromosomes and their associated Ran-GTP gradient and the presence of F-actin. Hence authors present here a potentially new mechanism explaining why paternal chromosomes are not extruded together with maternal ones in mouse zygotes.

Major comments:

1/ Authors insist on the presence of a so far uncharacterized transition zone, that they name (FA) which is ring-like and where Juno and CD9 accumulate a bit less than in the region opposite chromosomes. This (FA) region could correspond to the transition zone between the actin cap, devoid of microvilli, and the microvilli enriched cortex. Yet it is not clear in the work why they insist such much on this ring-like structure and what exactly its role is. Furthermore, they present data on this transition region only in unfertilized metaphase II arrested mouse oocytes (Figs 2, 3, 4, 5). Does this transition zone persist after fertilization (not obvious from images shown in Fig 6A)? If so, is it important in the process investigated here?

2/ Related to point 1, authors suggest that, in this region, Juno is more dynamic than in the microvilli dense cortical area (Fig 3C). Why is it important to show this? Is it also the case after fertilization? If authors perform FRAP on Juno do they indeed observe that it has a more important turn-over in this area compared to its turn-over in the microvilli dense cortical area?

3/ To further strengthened their work and make it a bit less descriptive, it would be nice if authors could artificially increase the concentration of Juno and/or CD9 in the cortical region above maternal chromosomes (maybe by targeting these proteins all-over the cortex?) and check whether this induces the fusion of the sperm closer to maternal chromosomes (less than 20 microm away from mom's chromosomes). Alternatively if authors overexpress RCC1 or RanQ69L, as done in (Dumont JCB 2007; Deng Dev Cell 2007), do they impact Juno and CD9 localization, as well as the formation of this transition zone? Do they impact the site of sperm fusion?

4/ Since authors cite Longo and Chen Dev Biol 1985, they should add in their citations the other papers referenced in the Introductory part of this review.

Reviewer #2 (Comments to the Authors (Required)):

This manuscript explores an important biological question: how mammalian oocytes manage to

finish meiosis II without eliminating the paternal chromosomes. Electron and fluorescence microscopy show a new ring-like structure surrounding the actin cap. The ring contains membrane receptors involved in sperm fusion but seems to use actin-mediated transport to move receptors away from the maternal genome. RanGTP and F-actin are involved in keeping the paternal chromosomes away from the maternal chromosomes, and F-actin also plays a role in moving the paternal chromosomes away from the maternal chromosomes. Finally, inability to keep the genomes apart until the extrusion of the second polar body increases the chance of eliminating the paternal genome in the polar body. Overall, the data are clearly presented and analyzed and of high quality, and the manuscript is well written.

Major points:

1. Figure 5D - While the control oocytes show almost no fusion at 30-60{degree sign} angle, LatB treatment seems to enrich for fusions around in area compared to other areas. How do the authors interpret this finding? Is there some additional, sperm-attracting signal in this area that is normally inhibited by F-actin?
2. The authors show that inhibition of F-actin disrupts the FA "ring-like" structure that seemingly transports Juno/CD9 proteins toward the DA zone, away from the maternal genome. While the authors mostly described the FA ring as a transitory route of Juno/CD9 structures toward the DA zone, it would be informative to estimate how often does sperm fusion occur within the FA zone as compared to simulated fusion for this region. One would expect less fusion due to relatively fewer Juno/CD9 structures. However, LatB-treated oocytes, which do not have a dynamic FA zone, show increased fusion events at 30-60{degree sign} degree angle. Related to comment 1 above, is it possible that natural fusion sites are enriched in the FA zone, and F-actin is required to minimize these fusion events?
3. Are there any consequences of forming one diploid pronucleus, which is one of the main consequences of placing the paternal chromosomes close to the maternal chromosomes (Figure 7)?
4. Discussion - paragraph 2 - "The disruption of F-actin significantly increased Juno/CD9 structures at regions proximal to maternal chromosomes, which suggests that F-actin is partially involved in the spatial control of Juno/CD9 structures". It is unclear how this statement is supported by the data. Figure 4 does not contain any quantitation of Juno/CD9 structures in LatB-treated oocytes, and the images do not clearly show an increase close to maternal chromosomes.

Minor points:

1. Figure 1B - A detailed explanation of the simulation can be found in the method section, but adding a one sentence explanation in the main text would facilitate the read.
2. Figure 1C - The trajectories are difficult to visualize and interpret. The plot on the left has many lines that overlap with each other. In both plots, it would be helpful to have information about time, possibly by color-coding the lines to indicate time progression. Also, what do the red arrowheads indicate?
3. Is there any bias in zona pellucida composition, which could affect the sperm fusion

independently of the membrane receptors and actin?

4. Figure 4A - spelling mistake "DSMO (FA)"

Reviewer #3 (Comments to the Authors (Required)):

This manuscript describes unprecedented live imaging of mouse fertilization and analysis of the position of the paternal DNA relative to the maternal DNA during meiosis II. The results presented in Figure 7 demonstrate the high significance of controlling the sperm fusion site. Sperm DNA injected adjacent to the meiotic spindle is frequently extruded in the second polar body which would result in a haploid (dead) embryo. This area is understudied due to technical limitations that have been partially overcome by these investigators. This work should definitely be published in JCB and will be of great interest to a wide audience of cell and developmental biologists. However, there are serious problems with the analysis presented as far as distinguishing whether fusion sites and paternal DNA movements in the zygote are actually directed away from the maternal DNA or if they are random. If fusion sites and paternal DNA movements are random rather than controlled, the study should still be published because it could be that the relative sizes of the egg and meiotic spindle are sufficient to generate the desired effect of keeping the paternal DNA and meiotic spindle apart. In addition to the outstanding Fig. 7 microinjection results, the staining pattern of Juno and CD9, proteins required for sperm egg fusion, in discrete zones that correlate with EM showing microvilli in different zones is particularly interesting and significant. The detailed comments below are meant to help make this a great paper.

Details

The first result of the paper concludes with "The distribution of egg-sperm fusion sites was significantly biased to regions distal to maternal chromosomes (Fig. 1B and S1C). These findings suggest that egg-sperm fusion is disfavored at cell surface regions in proximity to maternal chromosomes." This conclusion is not supported by the data shown in Fig. 1 and S1B (reference to S1C appears to be a typographical error). If the numbers over the bar graphs are p values, then 3 out of 4 of the values comparing observed vs random distribution are not significant. The p values in fig. 1B and S1B need to be labeled as p values and the statistical test used must be described. The model used to predict random distribution must also be described. A drawing explaining the angle between maternal chromosomes and sperm fusion site (like the one for distance in Fig. S1) should be added to fig. 1B. The authors need to describe this result in a more convincing way. The number of predicted and observed fusion events at 90-120 degrees is much higher than at 0-30 degrees or 150-180 degrees because the actual surface area of egg is much higher at 90-120 degrees. Presenting the values as fusion events/surface area might be a more convincing way to support the author's conclusion, however, n may not be high enough. Comparing 0-30 degrees (over the spindle) to 150-180 degrees (equal surface area at opposite pole is 0/79 vs 5/79 fusion events. Comparing these with fisher's exact test yields a not significant p value of 0.0586.

Analysis of the movement of the paternal DNA could also be better explained. It is not clear in the current presentation whether the paternal DNA moves preferentially away from the maternal DNA or if the movement is just random. The current analysis is again biased by the fact that the maternal DNA occupies a small volume compared to the whole egg. In Fig 1C(right) and Fig. 1E, only paternal DNA that fuses very close to the maternal DNA appears to move away from the maternal DNA and this could be random movement since there are more potential vectors away than toward

the maternal DNA.

One of the perceived limitations to live imaging of in vitro fertilization is the fact that fertilization requires the zona pelucida and the zona pelucida is typically removed before live imaging of mouse oocytes. The "culture and microinjection of mouse eggs" section of the materials and methods states that the zona pelucida was removed with collagenase. The "time lapse imaging of in vitro fertilization" section states that the zona pellucida was softened with glutathione and in some cases a hole was made with a piezo driven pipette. The first results section does not state which method was used. The legend for Fig. 1 does not state which method was used. Since manipulation of the zona pelucida could affect the site of fertilization, it is critical that for each set of experimental results, the exact manipulation of the zona pelucida is stated. The reason for each manipulation of the zona pelucida should also be clearly stated. The possibility that manipulation of the zona pelucida might affect the results should be discussed. The legend for Fig. 1 states that some data includes polyspermic zygotes whereas other data does not. The authors should clearly state the frequency of polyspermy for each set of data.

The sequence of the MBD construct used to label sperm in the cytoplasm of the egg needs to be provided either as a supplement or as an accession number in the Materials and Methods.

Formatting the manuscript with page numbers and line numbers would help reviewers write accurate critiques.

N (# of oocytes) is needed for the EM and kymograph results in Fig. 3.

The conclusion: "F-actin contributes to but is not solely responsible for blocking the formation of Juno/CD9 structures in proximity to maternal chromosomes" based on data in Fig. 4 depends completely on Fig.S2 which shows the extent of actin depolymerization after latrunculin treatment. Fig. S2 needs to show quantitation of fluorescence intensity from multiple oocytes. The legend of Fig. S2 (rather than just the materials and methods) should state that this is phalloidin staining. An image of a latrunculin-treated oocyte with the contrast enhanced should be shown to reveal whether cortical F-actin remains. The possibility of partial actin depolymerization after latrunculin treatment is also critical for the conclusion that the GTP-ran result is not simply due to disruption of the actin cap by ranT24N. To make this comparison, the authors should show quantitation of actin cap phalloidin staining intensity in control, latrunculin and ranT24N.

Much more detail is needed for the live imaging used to track paternal DNA location. Specifically, what was the exposure time, the number of focal planes captured, the time required to collect one z series and the time interval between images. This information should be in the materials and methods along with a discussion of the possibility that fusion occurs at 0-30 degrees just as frequently as at 150-180 degrees but that the paternal DNA is swept away from the 0-30 degree zone before it can be imaged.

Fig. 6B should indicate the maternal and paternal DNA with arrows.

In the section on tracking paternal DNA in latrunculin-treated zygotes, why is the denominator different (5/10 vs 11/21) ? There are no page or line numbers to refer to.

The conclusion: "These results suggest that F-actin dependent mechanisms keep paternal chromosomes away from maternal chromosomes after sperm fusion." Suggests that this is a different mechanism than the control of sperm fusion site. But there is only one example of a DMSO

control that fertilized closer than 30 μm from the maternal DNA and many in the latrunculin-treated zygotes. This conclusion should be stated in a way that makes it clear that latrunculin may only affect the fusion site rather than affecting movement after fertilization. Different data presentation/explanation is needed to support a role for F-actin in movement of paternal DNA after fusion.

June 14, 2021

Dr. Arshad Desai

Editor, Journal of Cell Biology

Dr. Dan Simon

Editor, Journal of Cell Biology

Dear Dr. Arshad Desai and Dr. Dan Simon,

Thank you very much for your decision letter dated January 14, 2021, giving us the opportunity to submit a revised copy of our manuscript entitled "RanGTP and F-actin prevent sperm fusion near the maternal spindle in mouse eggs" (202012001) for publication in the Journal of Cell Biology. We also would like to take this opportunity to express our gratitude to the Reviewers for their positive feedback and constructive comments to improve our manuscript.

We have revised our manuscript to address all of the Reviewers' comments as detailed on the following pages. We exchanged the order of Fig. 6 and Fig. 7, and incorporated new data in the modified Figures 1B, 1F, 3D, 3E, 4C, 7D, S1C, S2A, S3, S4 and S5 along with appropriate revisions in the text. Since these new data strengthen our model, we changed the title of our manuscript "RanGTP and the actin cytoskeleton keep paternal and maternal chromosomes apart during fertilization". All changes in the manuscript are highlighted in red. Our responses to each of the Reviewers' comments are on the following pages with their original comments in black and our response in blue.

We would like to ask for a reevaluation of our revised manuscript and we feel that the revisions made in response to the comments and suggestions of the Reviewers have significantly clarified and strengthened the manuscript. Hopefully, it is now suitable for publication in the *Journal of Cell Biology*.

Sincerely,

Masashi Mori

Email: masashi.mori@riken.jp

Masahito Ikawa,

Email: ikawa@biken.osaka-u.ac.jp

Editor Comments: Overall, the reviewers are enthusiastic about your study stating that it addresses an important question and they are complimentary of the quality of the data presented in the manuscript. We invite you to submit a revision if you can address the reviewers' key concerns, as outlined here.

Response: We were very happy to see that all 3 Reviewers appreciated the novelty and significance of our study and were in favor of publishing the manuscript after suitable revisions. We feel that we replied to all of their comments and hope that the Reviewers are satisfied with the revisions made and will now approve the manuscript for publication in the *Journal of Cell Biology*.

Reviewer #1 (Comments to the Authors (Required)):

Authors are addressing here an important question: how is the haploid set of paternal chromosomes not extruded together with the maternal set of separating chromosomes following fertilization of mouse metaphase II oocytes? This is a long-standing question that has so far been partly resolved by prior works showing:

(1) that maternal chromosomes induce a cortical differentiation, which corresponds to a region above them devoid of microvilli and enriched in actin, that does not favour sperm entry in proximity to maternal chromosomes (Maro J Emb Exp Morph 1984; Longo and Chen Dev Biol 1985). Similarly the paternal set of chromosomes also induces what has been referred as a fertilization cone, enriched in F-actin and devoid of microvilli (Shalgi Gamete Res 1978; Maro J Emb Exp Morph 1984; Longo and Chen Dev Biol 1985).

(2) that maternal chromosomes produce a Ran-GTP gradient potentially perturbed by the expression of Ran mutant forms (Dumont JCB 2007) that induces the above-mentioned cortical differentiation (Deng Dev Cell 2007) as well as a cytoplasmic flow that will tend to send sperm chromosomes away from maternal ones (Yi Nat Cell Biol 2011).

The novelty that authors are bringing here is that this cortical differentiated region is also devoid of two major players in sperm/egg membrane fusion, Juno and CD9. They show using beautiful and convincing live imaging that the localisation of these two proteins mimics that of microvilli, accumulating mostly where the protrusions are denser in the cortex of the oocyte. Using a series of perturbation experiments, authors also demonstrate that Juno and CD9 localisations depend on the presence of chromosomes and their associated Ran-GTP gradient and the presence of F-actin. Hence authors present here a potentially new mechanism explaining why paternal chromosomes are not extruded together with maternal ones in mouse zygotes.

Response: Thank you for your appreciation of the novelty and importance of our study and the impact of our results. We feel that the revisions made as detailed point-by-point below have significantly clarified and strengthened the text and we appreciate your constructive suggestions.

Major comments:

1/ Authors insist on the presence of a so far uncharacterized transition zone, that they name (FA) which is ring-like and where Juno and CD9 accumulate a bit less than in the region opposite chromosomes. This

(FA) region could correspond to the transition zone between the actin cap, devoid of microvilli, and the microvilli enriched cortex. Yet it is not clear in the work why they insist such much on this ring-like structure and what exactly its role is. Furthermore, they present data on this transition region only in unfertilized metaphase II arrested mouse oocytes (Figs 2, 3, 4, 5). Does this transition zone persist after fertilization (not obvious from images shown in Fig 6A)? If so, is it important in the process investigated here?

Response: Thank you for your appreciation of the importance of our study and we are also interested in the significance of the FA region. Since Juno structures move away from maternal chromosomes in the FA region, we hypothesized that when sperm bind to the FA region, they move on the egg surface before their fusion. We followed sperm movements by tracking the fluorescent sperm tails (Su9-DsRed2) and found that sperm which bound to the FA region move away from maternal chromosomes (Fig. 3D, E). On the other hand, sperm bound to the opposite half of maternal chromosomes did not move. These data suggest that the movement of Juno structures in the FA region functions to transfer bound sperm before their fusion (page 7, line 187). After fertilization, the FA region was also observed around the fertilization cone (see data below). The role of the FA region after fertilization is unknown. We did not expect that the FA region would contribute to the spatial separation of paternal and maternal chromosomes. Thank you for this helpful comment as it helped us to improve this study.

2/ Related to point 1, authors suggest that, in this region, Uno is more dynamic than in the microvilli dense cortical area (Fig 3C). Why is it important to show this? Is it also the case after fertilization? If authors perform FRAP on Juno do they indeed observe that it has a more important turn-over in this area compared to its turn-over in the microvilli dense cortical area?

Response: Our claim is that Juno structures, not Juno molecules, are more dynamic in the FA region than in the DA region. These structures in the FA region move away from maternal chromosomes (Movie 3). In the DA region, the tips of the Juno structures are moving, but their bases look stable (Movie 3) (page 7, line 185). Furthermore, we performed FRAP experiments to measure the turn-over rate of Juno molecules. The data show that Juno molecules are more dynamic in the FA region than in the DA region (see data below). However, since not only Juno molecules but also Juno structures are moving in the FA region, we cannot distinguish whether the recovery of Juno fluorescence is due to the dynamics of the Juno molecules or Juno structures.

3/ To further strengthened their work and make it a bit less descriptive, it would be nice if authors could artificially increase the concentration of Juno and/or CD9 in the cortical region above maternal chromosomes (maybe by targeting these proteins all-over the cortex?) and check whether this induces the fusion of the sperm closer to maternal chromosomes (less than 20 microm away from mom's chromosomes). Alternatively if authors overexpress RCC1 or RanQ69L, as done in (Dumont JCB 2007; Deng Dev Cell 2007), do they impact Juno and CD9 localization, as well as the formation of this transition zone? Do they impact the site of sperm fusion?

Response: We agree that it is important to perturb the localization of Juno structures. To increase the concentration of Juno structures in the cortical region above maternal chromosomes, we used Ran-inhibited eggs by injecting RanT24N. In those eggs, Juno structures were found all over the cell surface, including the cortical region above maternal chromosomes (Fig. 5B). Furthermore, sperm could fuse at the region proximal to maternal chromosomes in the Ran-inhibited eggs (Fig. 5C). We hope that these experiments sufficiently address your comment (page 8, line 222).

We have also tried RanGTP overexpression. We injected RanQ69L (the constitutively active form) protein as much as possible into unfertilized eggs (see data below). We observed that the chromosomes became swollen and the actin intensity on the actin cap was decreased. However, the localization of Juno structures did not change, and sperm fused as normal (sperm fusion sites also looked normal). Since it is difficult to judge how much the RanGTP level was increased in these overexpression experiments, we have not included this data in the paper.

4/ Since authors cite Longo and Chen, Dev Biol, 1985, they should add in their citations the other papers referenced in the Introductory part of this review.

Response: We have now cited the papers that showed actin filaments accumulate at the cortex above maternal chromosomes as a reference in the Introduction (page 3, line 69).

Reviewer #2 (Comments to the Authors (Required)):

This manuscript explores an important biological question: how mammalian oocytes manage to finish meiosis II without eliminating the paternal chromosomes. Electron and fluorescence microscopy show a new ring-like structure surrounding the actin cap. The ring contains membrane receptors involved in sperm fusion but seems to use actin-mediated transport to move receptors away from the maternal genome. RanGTP and F-actin are involved in keeping the paternal chromosomes away from the maternal chromosomes, and F-actin also plays a role in moving the paternal chromosomes away from the maternal chromosomes. Finally, inability to keep the genomes apart until the extrusion of the

second polar body increases the chance of eliminating the paternal genome in the polar body. Overall, the data are clearly presented and analyzed and of high quality, and the manuscript is well written.

Response: Thank you for your appreciation of the quality and importance of our study and the impact of our results. We feel that the revisions made as detailed point-by-point below have significantly clarified and strengthened the text and we appreciate your constructive suggestions.

Major points:

1. Figure 5D - While the control oocytes show almost no fusion at 30-60{degree sign} angle, LatB treatment seems to enrich for fusions around in area compared to other areas. How do the authors interpret this finding? Is there some additional, sperm-attracting signal in this area that is normally inhibited by F-actin?

Response: We tried to quantify the localization of Juno structures in LatB-treated eggs (Fig. 4C). In LatB-treated eggs, the localization patterns of Juno and CD9 in the FA region were disrupted, but Juno/CD9 structures were still absent around maternal chromosomes (Movie 5). The spot-like Juno/CD9 structures were translocated or appeared de novo in this former FA region (Fig. 4B, Movie 5). The border of Juno/CD9-defined segments was less clear but was roughly localized where the NA/FA border is normally formed in control eggs (Fig. 4C_Majority). Furthermore, a small population of the Juno/CD9 structures were localized even nearer to maternal chromosomes (Fig. 4C_closest). Thus, F-actin is required to block the formation of the spot-like Juno/CD9 structures in the FA region, which may contribute to the inhibition of sperm fusion in the FA region (page 7, line 203). On the other hand, we showed that the lamellipodia-like Juno/CD9 structures in the FA region have a directional movement and that sperm bound in the FA region move away from maternal chromosomes (Fig. 3C, D, E). These results suggested that F-actin may function to form and transfer the lamellipodia-like Juno/CD9 structures in the FA region to move the sperm away from maternal chromosomes before sperm fusion (page 7, line 183).

2. The authors show that inhibition of F-actin disrupts the FA "ring-like" structure that seemingly transports Juno/CD9 proteins toward the DA zone, away from the maternal genome. While the authors mostly described the FA ring as a transitory route of Juno/CD9 structures toward the DA zone, it would be informative to estimate how often does sperm fusion occur within the FA zone as compared to simulated fusion for this region. One would expect less fusion due to relatively fewer Juno/CD9 structures. However, LatB-treated oocytes, which do not have a dynamic FA zone, show increased fusion events at 30-60{degree sign} degree angle. Related to comment 1 above, is it possible that natural fusion sites are enriched in the FA zone, and F-actin is required to minimize these fusion events?

Response: We tried to observe where sperm bind on egg surface and found that sperm could bind the FA region (roughly 25-35 μm from maternal chromosomes) (Fig. 3F). In these experiments, since sperm penetrated from the holes in the zona pellucida near maternal spindles, it is difficult to estimate how often sperm bound within the FA region. Sperm bound in the FA region move toward the DA region before their fusion.

3. Are there any consequences of forming one diploid pronucleus, which is one of the main consequences of placing the paternal chromosomes close to the maternal chromosomes (Figure 7)?

Response: In our ICSI experiments, some embryos formed a single pronucleus. In humans, 1.6-7.7% of embryos contain a single pronucleus in assisted reproductive technology including ICSI. Some embryos with a single pronucleus can result in viable pregnancies with no apparent anomalies, but their overall success rate of blastocyst formation is lower than embryos with two pronuclei (Itoi et al., 2015).

According to our study, embryos with a single pronucleus likely form because paternal chromosomes are discarded into a polar body or are captured by maternal chromosomes in embryos (page12, line339). In mice, we expect that embryos with a single pronucleus show a lower success rate of blastocyst formation as in humans.

4. Discussion - paragraph 2 - "The disruption of F-actin significantly increased Juno/CD9 structures at regions proximal to maternal chromosomes, which suggests that F-actin is partially involved in the spatial control of Juno/CD9 structures". It is unclear how this statement is supported by the data. Figure 4 does not contain any quantitation of Juno/CD9 structures in LatB-treated oocytes, and the images do not clearly show an increase close to maternal chromosomes.

Response: We quantified the localization of Juno/CD9 structures in LatB-treated eggs (Fig. 4C) (page 7, line 203).

Minor points:

1. Figure 1B - A detailed explanation of the simulation can be found in the method section, but adding a one sentence explanation in the main text would facilitate the read.

Response: We added a sentence to explain the simulation (page 4, line 114), and an illustration in Fig. 1B.

2. Figure 1C - The trajectories are difficult to visualize and interpret. The plot on the left has many lines that overlap with each other. In both plots, it would be helpful to have information about time, possibly by color-coding the lines to indicate time progression. Also, what do the red arrowheads indicate?

Response: We made figures using fewer trajectories in Fig. S1. The red arrowheads in Fig. 1C and 1E indicate the timing of paternal chromosomes which fuse within the 30 μm region to start directionally moving away from maternal chromosomes.

3. Is there any bias in zona pellucida composition, which could affect the sperm fusion independently of the membrane receptors and actin?

Response: We measured the mechanical property by the special microneedle-based setup and found the stiffness of zona pellucida is uniform. Furthermore, we measured the porosity by filling its porous space with fluorescent dextran and found the porosity of zona pellucida is also uniform along the egg surface independent of the position of maternal chromosomes (Fig. S3 and Fig. S4, page6, line149). We also tried to observe ZP proteins which are the main components of zona pellucida (data of ZP3 below). Since their localization pattern changed dependent on the fixation conditions, we have not included this data in the paper.

4. Figure 4A - spelling mistake "DSMO (FA)"

Response: We corrected the spelling of DMSO in Fig. 4A, thanks for noticing that.

Reviewer #3 (Comments to the Authors (Required)):

This manuscript describes unprecedented live imaging of mouse fertilization and analysis of the position of the paternal DNA relative to the maternal DNA during meiosis II. The results presented in Figure 7 demonstrate the high significance of controlling the sperm fusion site. Sperm DNA injected adjacent to the meiotic spindle is frequently extruded in the second polar body which would result in a haploid (dead) embryo. This area is understudied due to technical limitations that have been partially overcome by these investigators. This work should definitely be published in JCB and will be of great interest to a wide audience of cell and developmental biologists. However, there are serious problems with the analysis presented as far as distinguishing whether fusion sites and paternal DNA movements in the zygote are actually directed away from the maternal DNA or if they are random. If fusion sites and paternal DNA movements are random rather than controlled, the study should still be published because it could be that the relative sizes of the egg and meiotic spindle are sufficient to generate the desired effect of keeping the paternal DNA and meiotic spindle apart. In addition to the outstanding Fig. 7 microinjection results, the staining pattern of Juno and CD9, proteins required for sperm egg fusion, in discrete zones that correlate with EM showing microvilli in different zones is particularly interesting and significant. The detailed comments below are meant to help make this a great paper.

Response: Thank you for your appreciation of the technological advances and importance of our study and the significant impact of our results in this field. We feel that the revisions made as detailed point-by-point below have significantly clarified and strengthened the text and we appreciate your constructive comments and suggestions.

Details

The first result of the paper concludes with "The distribution of egg-sperm fusion sites was significantly biased to regions distal to maternal chromosomes (Fig. 1B and S1C). These findings suggest that egg-sperm fusion is unfavored at cell surface regions in proximity to maternal chromosomes." This

conclusion is not supported by the data shown in Fig. 1 and S1B (reference to S1C appears to be a typographical error). If the numbers over the bar graphs are p values, then 3 out of 4 of the values comparing observed vs random distribution are not significant. The p values in fig. 1B and S1B need to be labeled as p values and the statistical test used must be described. The model used to predict random distribution must also be described. A drawing explaining the angle between maternal chromosomes and sperm fusion site (like the one for distance in Fig. S1) should be added to fig. 1B. The authors need to describe this result in a more convincing way. The number of predicted and observed fusion events at 90-120 degrees is much higher than at 0-30 degrees or 150-180 degrees because the actual surface area of egg is much higher at 90-120 degrees. Presenting the values as fusion events/surface area might be a more convincing way to support the author's conclusion, however, n may not be high enough. Comparing 0-30 degrees (over the spindle) to 150-180 degrees (equal surface area at opposite pole is 0/79 vs 5/79 fusion events. Comparing these with fisher's exact test yields a not significant p value of 0.0586.

Response: We agree with this comment and should have written that section more correctly. In the revised manuscript, we have added more data sets that support a significant bias in sperm fusion sites (Fig. 1B and S1B). Fisher's exact tests were used for Fig. 1B and Fig. S1B using expected assignment probability to each bin resulted from the simulation as a null hypothesis. Holm correction was used for the correction of multiple comparison. The statistical information is now included in the Figure Legend (Fig. 1B, page 24, line 684).

Analysis of the movement of the paternal DNA could also be better explained. It is not clear in the current presentation whether the paternal DNA moves preferentially away from the maternal DNA or if the movement is just random. The current analysis is again biased by the fact that the maternal DNA occupies a small volume compared to the whole egg. In Fig 1C(right) and Fig. 1E, only paternal DNA that fuses very close to the maternal DNA appears to move away from the maternal DNA and this could be random movement since there are more potential vectors away than toward the maternal DNA.

Response: We calculated the angular displacement of paternal chromosomes toward maternal chromosomes at every time point between sperm fusion and 100 min after anaphase onset. We tested 4 models to explain the movement of paternal chromosomes: the angular displacement has a correlation with the position of paternal chromosomes (Angle Only), with the time after sperm fusion (Time Only), with both these variables (Angle + Time) and with both these variables and their interaction (Angle x Time) (Fig. 1F and Fig. S2A). These models were evaluated based on the Akaike's Information Criterion values, and the most likely model was Angle x Time. According to the prediction of the Angle x Time model, we separated the data into the spindle half and the opposite half of zygotes and found that paternal chromosomes moved away from maternal chromosomes in the spindle half during an early phase of the fertilization process (Fig. 1F Right). These findings suggest that zygotes actively keep paternal chromosomes at a distance from maternal chromosomes after sperm fusion (page 5, line 127). Thank you for this great comment, which significantly improved the clarify and impact of this study.

One of the perceived limitations to live imaging of in vitro fertilization is the fact that fertilization requires the zona pelucida and the zona pelucida is typically removed before live imaging of mouse oocytes. The "culture and microinjection of mouse eggs" section of the materials and methods states that the zona pelucida was removed with collagenase. The "time lapse imaging of in vitro fertilization" section states that the zona pellucida was softened with glutathione and in some cases a hole was made with a piezo driven pipette. The first results section does not state which method was used. The legend for Fig. 1 does not state which method was used. Since manipulation of the zona pelucida could affect the site of fertilization, it is critical that for each set of experimental results, the exact manipulation of

the zona pelucida is stated. The reason for each manipulation of the zona pelucida should also be clearly stated. The possibility that manipulation of the zona pelucida might affect the results should be discussed. The legend for Fig. 1 states that some data includes polyspermic zygotes whereas other data does not. The authors should clearly state the frequency of polyspermy for each set of data.

Response: Thank you for these comments and suggestions and we agree that it is important to clarify these points in the manuscript. In this paper, we used 4 different conditions of the zona pellucida.

1, The zona pellucida was softened by treatment with glutathione (Fig. 1, Fig. 5C and Fig. 7). Since the zona pellucida was expanded evenly, the penetration site was likely random. However, it is difficult to control how soft the zona pellucida became, and sometimes sperm cannot penetrate the zona pellucida at all. Thus, we used this condition to observe the distribution of the sperm fusion site.

2, Single or two holes were made in the zona pellucida using a piezo-driven pipette without glutathione treatment (single hole in Fig. 5D, 6A, 6B and 6C, and two holes in Fig. 3D and 3F). Since we made holes near the perivitelline space that was often formed at the outer surface of the maternal spindle, the penetration site of sperm has a bias. At the beginning, we made holes independent of the maternal chromosomes to get more fertilized eggs. However, we noticed that the sperm tended to fuse in the spindle half after several experiments. Thus, we used this condition to obtain embryos in which sperm binds to the FA region.

3, The zona pellucida was removed by treatment with collagenase (Fig. 2, Fig. 3, Fig. 4, Fig. 5A). Since vesicles consisting of Juno and CD9 were observed in the perivitelline space, we removed the zona pellucida to image Juno and CD9 structures on the egg surface.

4, The zona pellucida was intact (Fig. 5B). Since it is difficult to inject eggs without the zona pellucida, we used this condition to observe Juno and CD9 structures after Ran protein injection.

The conditions of the zona pellucida and the polyspermy rate are now specified in the revised text and Figure Legends.

The sequence of the MBD construct used to label sperm in the cytoplasm of the egg needs to be provided either as a supplement or as an accession number in the Materials and Methods.

Response: The MBD construct has not yet been deposited. The design of the construct is described in a published paper (Yamagata et al, 2005 in references) by Kazuo Yamagata (yamagata@waka.kindai.ac.jp).

Formatting the manuscript with page numbers and line numbers would help reviewers write accurate critiques.

Response: We added page numbers and line numbers in the revised manuscript as requested.

N (# of oocytes) is needed for the EM and kymograph results in Fig. 3.

Response: Those numbers are now listed in the revised Figure Legend (Fig. 3A, C). EM: n=21; Kymograph: n=15

The conclusion: "F-actin contributes to but is not solely responsible for blocking the formation of Juno/CD9 structures in proximity to maternal chromosomes" based on data in Fig. 4 depends completely on Fig.S2 which shows the extent of actin depolymerization after latrunculin treatment. Fig. S2 needs to show quantitation of fluorescence intensity from multiple oocytes. The legend of Fig. S2 (rather than just the materials and methods) should state that this is phalloidin staining. An image of a latrunculin-treated oocyte with the contrast enhanced should be shown to reveal whether cortical F-actin remains. The possibility of partial actin depolymerization after latrunculin treatment is also critical for the conclusion that the GTP-ran result is not simply due to disruption of the actin cap by ranT24N. To make

this comparison, the authors should show quantitation of actin cap phalloidin staining intensity in control, latrunculin and ranT24N.

Response: Thank you very much for this comment. We agree that if actin structures remained at the actin cap after LatB treatment, there is the possibility that actin solely can inhibit sperm fusion downstream of RanGTP. We measured the F-actin intensity of microvilli area (Distal) and the actin cap (Proximal) in the DMSO or LatB treated eggs (Fig. S5A). The actin cortex was reduced at both Distal and Proximal areas in LatB-treated eggs, and the intensity at Proximal area was higher than that at Distal area. On the other hand, we measured the F-actin intensity also in Ran-inhibited eggs (Fig. S5B). The actin cortex was reduced only at Proximal area, but the intensity at Proximal area was still higher than that at Distal area. These data indicates that some stable actin cortex was remained around maternal chromosomes in LatB- treated eggs and Ran-inhibited eggs. Thus, even though some stable actin structures remained at the actin cap in both cases, sperm fused near maternal chromosomes in Ran-inhibited eggs, but not in LatB-treated eggs. These data support the conclusion that a stable F-actin structure at the actin cap is not the critical factor to block sperm fusion in LatB-treated eggs or in Ran inhibited eggs.

Much more detail is needed for the live imaging used to track paternal DNA location. Specifically, what was the exposure time, the number of focal planes captured, the time required to collect one z series and the time interval between images. This information should be in the materials and methods along with a discussion of the possibility that fusion occurs at 0-30 degrees just as frequently as at 150-180 degrees but that the paternal DNA is swept away from the 0-30 degree zone before it can be imaged.

Response: To track chromosomes, images were typically acquired at a 100–120 μm square image plane, a spatial resolution of 3 μm confocal sections covering 81–96 μm and a temporal resolution of 2.5 or 3 min with a confocal microscope. Since paternal chromosomes usually expand where sperm fuse and it takes much longer than the interval time of imaging (Fig. S1A), it is unlikely that we missed imaging sperm fusion near maternal chromosomes. We often recorded 10-30 eggs in parallel (page14, line390).

The conclusion: "These results suggest that F-actin dependent mechanisms keep paternal chromosomes away from maternal chromosomes after sperm fusion." Suggests that this is a different mechanism than the control of sperm fusion site. But there is only one example of a DMSO control that fertilized closer than 30 μm from the maternal DNA and many in the latrunculin-treated zygotes. This conclusion should be stated in a way that makes it clear that latrunculin may only affect the fusion site rather than affecting movement after fertilization. Different data presentation/explanation is needed to support a role for F-actin in movement of paternal DNA after fusion.

Response: Thank you for this comment. We struggled with this point for a long time. When we injected sperm heads near the spindle and plasma membrane, the paternal chromosomes often moved away from maternal chromosomes (Fig. 6A, move away). Thus, to analyze the behavior of paternal chromosomes following egg–sperm fusion in proximity to maternal chromosomes, we placed sperm heads underneath the NA region (Fig. 7D). In control eggs, a substantial population of paternal chromosomes moved away from maternal chromosomes (11/22), while others were captured by maternal chromosomes. In contrast, in LatB-treated eggs, we never observed paternal chromosomes moving away from maternal chromosomes (0/16). These results suggest that an F-actin-dependent mechanism acts to separate paternal chromosomes away from maternal chromosomes following fertilization. This mechanism is required but not sufficient to guarantee the protection of paternal chromosomes in cases where egg–sperm fusion occurs in proximity to maternal chromosomes (page 10, line 266).

July 13, 2021

RE: JCB Manuscript #202012001R

Dr. Masashi Mori
RIKEN Center for Developmental Biology
2-2-3 Minat ojima-minamimachi, Chuou-ku,
Kobe, Hyogo 650-0047
Japan

Dear Dr. Mori,

Thank you for submitting your revised manuscript entitled "RanGTP and the actin cytoskeleton keep paternal and maternal chromosomes apart during fertilization." We would be happy to publish your paper in JCB pending final revisions necessary to address remaining minor points from reviewers and to meet our formatting guidelines (see details below).

A. MANUSCRIPT ORGANIZATION AND FORMATTING:

Full guidelines are available on our Instructions for Authors page, <https://jcb.rupress.org/submission-guidelines#revised>. **Submission of a paper that does not conform to JCB guidelines will delay the acceptance of your manuscript.**

- 1) Text limits: Character count for Articles is < 40,000, not including spaces. Count includes title page, abstract, introduction, results, discussion, and acknowledgments. Count does not include materials and methods, figure legends, references, tables, or supplemental legends.
- 2) Figures limits: Articles may have up to 10 main text figures.
- 3) Figure formatting: Scale bars must be present on all microscopy images, including inset magnifications. Molecular weight or nucleic acid size markers must be included on all gel electrophoresis.
- 4) Statistical analysis: Error bars on graphic representations of numerical data must be clearly described in the figure legend. The number of independent data points (n) represented in a graph must be indicated in the legend. Statistical methods should be explained in full in the materials and methods. For figures presenting pooled data the statistical measure should be defined in the figure legends. Please also be sure to indicate the statistical tests used in each of your experiments (both in the figure legend itself and in a separate methods section) as well as the parameters of the test (for example, if you ran a t-test, please indicate if it was one- or two-sided, etc.). Also, if you used parametric tests, please indicate if the data distribution was tested for normality (and if so, how). If not, you must state something to the effect that "Data distribution was assumed to be normal but this was not formally tested."

- 5) Materials and methods: Should be comprehensive and not simply reference a previous publication for details on how an experiment was performed. Please provide full descriptions (at least in brief) in the text for readers who may not have access to referenced manuscripts. The text should not refer to methods "...as previously described."
- 6) For all cell lines, vectors, constructs/cDNAs, etc. - all genetic material: please include database / vendor ID (e.g., Addgene, ATCC, etc.) or if unavailable, please briefly describe their basic genetic features, even if described in other published work or gifted to you by other investigators. Please be sure to provide the sequences for all of your oligos: primers, si/shRNA, RNAi, gRNAs, etc. in the materials and methods. You must also indicate in the methods the source, species, and catalog numbers/vendor identifiers (where appropriate) for all of your antibodies, including secondary.
- 7) Microscope image acquisition: The following information must be provided about the acquisition and processing of images:
- Make and model of microscope
 - Type, magnification, and numerical aperture of the objective lenses
 - Temperature
 - Imaging medium
 - Fluorochromes
 - Camera make and model
 - Acquisition software
 - Any software used for image processing subsequent to data acquisition. Please include details and types of operations involved (e.g., type of deconvolution, 3D reconstitutions, surface or volume rendering, gamma adjustments, etc.).
- 8) References: There is no limit to the number of references cited in a manuscript. References should be cited parenthetically in the text by author and year of publication. Abbreviate the names of journals according to PubMed.
- 9) Supplemental materials: There are strict limits on the allowable amount of supplemental data. Articles may have up to 5 supplemental figures and 10 videos. Please also note that tables, like figures, should be provided as individual, editable files. A summary of all supplemental material should appear at the end of the Materials and methods section. Please include one brief sentence per item.
- 10) eTOC summary: A ~40-50 word summary that describes the context and significance of the findings for a general readership should be included on the title page. The statement should be written in the present tense and refer to the work in the third person. It should begin with "First author name(s) et al..." to match our preferred style.
- 11) Conflict of interest statement: JCB requires inclusion of a statement in the acknowledgements regarding competing financial interests. If no competing financial interests exist, please include the following statement: "The authors declare no competing financial interests." If competing interests are declared, please follow your statement of these competing interests with the following statement: "The authors declare no further competing financial interests."
- 12) A separate author contribution section is required following the Acknowledgments in all research manuscripts. All authors should be mentioned and designated by their first and middle initials and full surnames. We encourage use of the CRediT nomenclature (<https://casrai.org/credit/>).

13) ORCID IDs: ORCID IDs are unique identifiers allowing researchers to create a record of their various scholarly contributions in a single place. At resubmission of your final files, please consider providing an ORCID ID for as many contributing authors as possible.

B. FINAL FILES:

Thank you for this interesting contribution, we look forward to publishing your paper in Journal of Cell Biology.

Sincerely,

Arshad Desai, PhD
Monitoring Editor
Journal of Cell Biology

Dan Simon, PhD
Scientific Editor
Journal of Cell Biology

Reviewer #1 (Comments to the Authors (Required)):

I think it is important that authors systematically provide in the text, legend to figures and maybe also more clearly on figures (with stars or indication of ns) the p values that have been measured between a control and any treatment. In the absence of such p values, it is quite difficult to know if some experiments are conclusive or not (ex: Fig 2B, 4C, 5C, 5D).

Reviewer #2 (Comments to the Authors (Required)):

The manuscript is improved, but additional textual changes would help clarify the findings:

1. Figure 4C and Lines 205-207: "These Juno/CD9 structures often reached positions where the NA/FA border is normally found in control eggs (Fig. 4C, LatB_DA), and a small population of the Juno/CD9 structures were localized even nearer to maternal chromosomes (Fig. 4C, LatB_closest)". It is not clear how the measurement was performed for this figure. The authors should either provide a schematic for all the conditions or indicate on Fig.4 A/B which structures they compared. Since the FA ring is not visible upon LatB treatment (Fig. 4B), it is not clear what "LatB_DA" means and where it ends.

2. Fig. 5D: Based on the updated method section, it seems that the bias in fusion events at angle 30-60{degree sign} upon LatB treatment is due to the authors poking holes in the zona near the maternal chromosomes, rather than increased affinity of sperm to that area. If that is correct, the authors should explain the method of altering the zona in the text rather than the method section. This is crucial to understand the figures and their interpretation.

Reviewer #3 (Comments to the Authors (Required)):

This much improved manuscript demonstrates multiple mechanisms that contribute to keeping the maternal and paternal chromosomes of the mouse zygote apart until after extrusion of the second polar body. The sperm/egg fusion proteins, Juno and CD9, are excluded from the region over the MII spindle by a GTP-ran signal from maternal chromosomes. Surface-bound sperm move away from the spindle and paternal chromosomes after fusion move through the cytoplasm away from the meiotic spindle. In addition, injection of sperm adjacent to the spindle causes expulsion of paternal chromosomes into the second polar body. This last result demonstrates the significance of these pathways. This will be a seminal paper in a very understudied but highly significant field. Below are some very minor suggestions but the manuscript should be published in JCB.

Minor:

The units for the values on the x and y axes of Fig. 1C, the y axis of Fig. 1F and the y axis of Fig. 6B right should be more clearly shown. The only explanation is in the legend of Fig. 1: "1 unit shows the length between the sperm fusion site and the center of the egg". Length would have units of microns whereas "angular displacement" might have units of degrees.

Graphs showing "angles" should be labeled with "degrees" as the units. Fig. 1B, 2B, 4C, 5C, 5D, etc.

The description of the capture of paternal chromosomes by maternal chromosomes is a bit confusing. It seems more likely that the paternal chromosomes are captured by the spindle microtubules or by the acto-myosin polar body rather than by the maternal chromosomes. It is OK with this reviewer to leave the language as it is because it would require much more work to figure out the exact mechanism of "capture".

The statement in the last paragraph that the MII spindle is not always adjacent to the first polar body could have some citations because this has been addressed for human oocytes.

2nd Revision - Authors' Response to Reviewers: July 21, 2021

Dr. Arshad Desai

Editor, Journal of Cell Biology

Dr. Dan Simon

Editor, Journal of Cell Biology

Dear Dr. Arshad Desai and Dr. Dan Simon,

Thank you very much for your decision letter dated July 13, 2021, giving us the opportunity to publish our paper entitled " RanGTP and the actin cytoskeleton keep paternal and maternal chromosomes apart during fertilization " (202012001) in the Journal of Cell Biology. We also would like to express our gratitude to the Reviewers for more feedback and constructive comments to improve our manuscript.

We have revised our manuscript to address the Reviewers' comments as detailed on the following pages. We modified figure 1B, 1C, 1F, 2B, 4C, 5C, 5D, 6B, S1B and S5C. All changes in the manuscript are highlighted in **red**. Our responses to each of the Reviewers' comments are on the following pages with their original comments in black and our response in **blue**.

We hope that the revisions made in response to the comments and suggestions of the Reviewers have significantly clarified.

Sincerely,

Masashi Mori

Email: masashi.mori@riken.jp

Masahito Ikawa,

Email: ikawa@biken.osaka-u.ac.jp

Reviewer #1 (Comments to the Authors (Required)):

I think it is important that authors systematically provide in the text, legend to figures and maybe also more clearly on figures (with stars or indication of ns) the p values that have been measured between a control and any treatment. In the absence of such p values, it is quite difficult to know if some experiments are conclusive or not (ex: Fig 2B, 4C, 5C, 5D).

Response: We have modified Fig1B, Fig1F, Fig4C, Fig5D, FigS1B and FigS5C, which now show asterisks and 'ns', as well as their legends indicating the p values. Since we measured but did not compare the angles of FA and DA borders in Fig2B, we do not perform t.test.

Reviewer #2 (Comments to the Authors (Required)):

The manuscript is improved, but additional textual changes would help clarify the findings:

1. Figure 4C and Lines 205-207: "These Juno/CD9 structures often reached positions where the NA/FA border is normally found in control eggs (Fig. 4C, LatB_DA), and a small population of the Juno/CD9 structures were localized even nearer to maternal chromosomes (Fig. 4C, LatB_closest)". It is not clear how the measurement was performed for this figure. The authors should either provide a schematic for all the conditions or indicate on Fig.4 A/B which structures they compared. Since the FA ring is not visible upon LatB treatment (Fig. 4B), it is not clear what "LatB_DA" means and where it ends.

Response: We agree that it is important to clearly state how we defined the NA/DA border after LatB treatment, since the FA ring is not visible in these eggs. To define the border, we plotted the signal intensities of CD9 along a line on the maximum z-projection image, and the resultant plot (shown as Line plot profile) was used to acquire the position of local intensity increase (plot profile). The position was used to calculate the angle with the center of the egg in 3D. We acquired such angles from three lines per one oocyte and averaged them. This explanation has been included in the legend of Figure 4C.

2. Fig. 5D: Based on the updated method section, it seems that the bias in fusion events at angle 30-60{degree sign} upon LatB treatment is due to the authors poking holes in the zona near the maternal chromosomes, rather than increased affinity of sperm to that area. If that is correct, the authors should explain the method of altering the zona in the text rather than the method section. This is crucial to understand the figures and their interpretation.

Response: Thank you for pointing this out. We indeed observed a bias in fusion events at angles of 30-60 degrees in LatB-treated eggs, which is possibly due to the holes that were made in the zona. As you suggest, the revised manuscript now states, "Note that sperm fusion sites in LatB-treated eggs appeared to be enriched around angles of 30–60° rather than uniformly distributed at angles of 30–180°, which may be due to the holes that were made in the

zona pellucida near the perivitelline space in order to prevent polyspermy." in the main text (page9, line238).

Reviewer #3 (Comments to the Authors (Required)):

This much improved manuscript demonstrates multiple mechanisms that contribute to keeping the maternal and paternal chromosomes of the mouse zygote apart until after extrusion of the second polar body. The sperm/egg fusion proteins, Juno and CD9, are excluded from the region over the MII spindle by a GTP-ran signal from maternal chromosomes. Surface-bound sperm move away from the spindle and paternal chromosomes after fusion move through the cytoplasm away from the meiotic spindle. In addition, injection of sperm adjacent to the spindle causes expulsion of paternal chromosomes into the second polar body. This last result demonstrates the significance of these pathways. This will be a seminal paper in a very understudied but highly significant field. Below are some very minor suggestions but the manuscript should be published in JCB.

Minor:

The units for the values on the x and y axes of Fig. 1C, the y axis of Fig. 1F and the y axis of Fig. 6B right should be more clearly shown. The only explanation is in the legend of Fig. 1: "1 unit shows the length between the sperm fusion site and the center of the egg". Length would have units of microns whereas "angular displacement" might have units of degrees.

Response: The plots in Figure 1C and 6B show normalized values. In Figure 1C, the values on the x and y axes were normalized by the radius of the egg. In Figure 6B, the values were normalized by the distance from the center of the egg to the maternal chromosomes. The plot in Fig. 1F shows angular displacement (degree per minute) and angle (degree). We now appropriately label the axes of the plots in the figures and state information in the legends.

Graphs showing "angles" should be labeled with "degrees" as the units. Fig. 1B, 2B, 4C, 5C, 5D, etc.

Response: We added units in Fig1B, 1F, 2B, 4C, 5C, 5D, S1B, S2A.

The description of the capture of paternal chromosomes by maternal chromosomes is a bit confusing. It seems more likely that the paternal chromosomes are captured by the spindle microtubules or by the acto-myosin polar body rather than by the maternal chromosomes. It is OK with this reviewer to leave the language as it is because it would require much more work to figure out the exact mechanism of "capture".

Response: Thank you for this comment. We have changed the wording and now use "fused" (page9, line252 and page10, line269).

The statement in the last paragraph that the MII spindle is not always adjacent to the first polar body could have some citations because this has been addressed for human oocytes.

Response: We have now cited the papers that showed the polar body does not always mark the position of maternal chromosomes in the last paragraph (page12, line352).